# ROAM: A RELATION-AWARE OPTIMAL TRANSPORT-BASED ADAPTIVE MIXTURE-OF-EXPERT-GROUP FRAMEWORK FOR MULTIMODAL KNOWLEDGE GRAPH COMPLETION

## ABSTRACT

Multimodal Knowledge Graph Completion (MMKGC) aims to predict missing facts by reasoning over heterogeneous information sources, including structural, textual, and visual modalities. A central challenge in this task lies in effectively integrating modality-specific information while preserving their distinct semantics and mitigating cross-modal interference. Recent efforts have explored employing Mixture-of-Experts (MoE) architectures to address this issue. However, many of these approaches rely on predefined expert routing or task-agnostic distance measures, thereby limiting their adaptability and performance. This paper proposes **ROAM**, a Relation-aware Optimal Transport-based Adaptive Mixture-of-Expert-Group framework, which dynamically routes modality-specific embeddings across expert groups conditioned on relation semantics. Specifically, *ROAM* first establishes modality-specialized expert groups to disentangle representation learning across modalities. Then, an Optimal Transport-based gating mechanism is introduced with a learnable, relation-conditioned cost function. Expert groups are further represented dynamically via their constituent parameters, enabling context-sensitive routing and capturing relation-aware specialization. Extensive experiments on multiple MMKGC benchmarks demonstrate that *ROAM* achieves state-of-the-art performance, achieving up to 9.76% relative gains.

## 1 INTRODUCTION

Multimodal Knowledge Graphs (MMKGs) extend traditional knowledge graphs by enriching structural triples with auxiliary modality information such as images, textual descriptions, and other sensory data. By incorporating cross-modal signals, MMKGs enable a broad range of applications such as visual question answering (Dong et al., 2024), multimodal reasoning (Lee et al., 2024), recommendation (Liang et al., 2024), and *etc*. To support these downstream applications, the task of **Multimodal Knowledge Graph Completion (MMKGC)** has emerged, which aims to predict missing en-

Table 1: Comparison of existing methods that integrate MoE and/or OT across various multi-modal tasks. Symbol categories: ◇: unconditional cost function, ♦: relation-conditioned cost, △: predetermined cost, ▲: learnable cost, ♡: prototype-based expert representation, ♠: parameter-driven expert representation (ours).

| Method | MoE | OT | Design Features |
|---|---|---|---|
| Sinkhorn-MoE (Clark et al., 2022) | ✓ | ✓ | ◇ △ ♡ |
| SVMoE (Vesaghati & Zareapoor, 2024) | ✓ | ✓ | ◇ △ ♡ |
| FedOTP (Liu et al., 2024) | ✓ | ✓ | ◇ △ ♡ |
| MoMoK (Zhang et al., 2025) | ✓ | × | ♡ |
| ChoicE (Xue et al., 2025) | ✓ | × | ♡ |
| MG-VMoE (Zhou et al., 2025) | ✓ | × | ♡ |
| OTKGE (Cao et al., 2022) | × | ✓ | ◇ △ |
| OT-MEL (Zhang et al., 2024d) | × | ✓ | ◇ △ |
| OTGNET (Lan et al., 2025) | × | ✓ | ◇ △ |
| OTMKGRL (Wang & Shen, 2025) | × | ✓ | ◇ △ |
| *ROAM* (Proposed) | ✓ | ✓ | ♦ ▲ ♠ |

tity links by jointly modeling structural and modality-aware representations. To this end, various methods have been proposed, including modality alignment (Lu et al., 2024), relation-aware fusion strategies (Shang et al., 2024), and contrastive representation learning (Zhu et al., 2025).

Among recent efforts, two promising directions have attracted attention: **Mixture-of-Experts (MoE)** and **Optimal Transport (OT)**. MoE-based approaches promote specialization by routing inputs to a subset of modality-specific experts, enabling adaptive representation learning. These models have been applied to MMKGC via customized expert selection (Zhang et al., 2025), adaptive

routing across encoding and decoding (Xue et al., 2025), and variational expert modeling for multi-modal extraction (Zhou et al., 2025). On the other hand, OT-based methods approach MMKGC as distributional alignment, leveraging transport-based strategies for modality matching (Zhang et al., 2024d), multimodal link prediction (Lan et al., 2025), and cross-modal representation learning (Cao et al., 2022; Wang & Shen, 2025). Another promising direction involves integrating OT with MoE for cost-aware expert routing (Liu et al., 2023b; Vesaghati & Zareapoor, 2024).

While MoE and OT have shown strong potential in multimodal learning, their joint application to MMKGC remains unexplored. *To the best of our knowledge, no prior work has unified OT and MoE for the MMKGC task.* Moreover, directly applying OT-based MoE approaches to MMKGC present **key challenges**. *First*, relation context is essential for guiding entity reasoning, yet current OT-MoE designs are relation-agnostic, limiting their capacity for fine-grained relational modeling. *Second*, methods typically compute transport costs using geometric distances or input-expert similarities, without incorporating downstream objectives. *Third*, experts are often represented by predefined prototype vectors, which fail to reflect their learning behavior during model fine-tuning.

To address the aforementioned challenges, we propose the Relation-aware Optimal transport-based Adaptive Mixture-of-Expert-Group (**ROAM**) algorithm. *ROAM* comprises two core modules: **Modality-Specific Expert Group (MoSEG)** and **Relation-Guided Optimal Transport Routing with learnable cost (ReGOR)**. Specifically, MoSEG organizes experts into groups, where each group contains modality-specific experts. Within each group, the outputs of experts are aggregated via a relation-aware gating mechanism, allowing the model to capture semantic variations across different relations. ReGOR then performs group selection via Optimal Transport, where the transport cost is a learnable function conditioned on relation semantics. Rather than relying on traditional prototypes, group representations are dynamically constructed from constituent expert parameters, enabling adaptive and relation-sensitive routing. A comparison between existing and the proposed method is presented in Table 1 for convenience. Our key contributions are summarized as follows:

- We propose *ROAM*, a novel framework for Multimodal Knowledge Graph Completion (MMKGC) that incorporates relation-aware expert group routing.
- *ROAM* introduces an OT-based gating mechanism with a learnable, relation-conditioned cost function, enabling dynamic and semantically aligned selection.
- Group representations are constructed from parameters of constituent modality-specific experts, capturing adaptive specialization without relying on prototypes.
- Experiments on four MMKGC benchmarks show that *ROAM* consistently outperforms state-of-the-arts[1].

## 2 RELATED WORK

**Multimodal Knowledge Graph Completion (MMKGC)** aims to predict missing entities or relations in knowledge graphs by leveraging both structural triples and auxiliary modality data. Most existing approaches adopt a two-stage paradigm: modality-specific encoders extract representations, *e.g.*, VGG (Simonyan & Zisserman, 2015) for images, BERT (Devlin et al., 2019) for texts, which are then fused via attention (Shang et al., 2024), contrastive learning (Lu et al., 2024; Zhao et al., 2024), or collaborative mechanisms (Zhu et al., 2025).

Recent work explores **Mixture-of-Experts (MoE)** architectures to enable adaptive and modular fusion. Specifically, MoE decomposes a large model into $K$ experts, with a gating function dynamically selecting a subset for each input ($x$): $y = \sum_{k=1}^{K} G_k(x) \cdot E_k(x)$, where $G_k(x)$ is the gating weight and $E_k(x)$ is the output of the $k$-th expert. **VL-MoE** (Shen et al., 2023) employs visual and textual experts, where image and text tokens are routed to relevant experts via sparse gating. **MOME** (Shen et al., 2024) introduces dual expert modules for adaptive visual fusion and task-aware language specialization. **MOAI** (Lee et al., 2025) employs conditional softmax gating to fuse image and text features, while **MoMoK** (Zhang et al., 2025) incorporates relation-guided expert selection through mutual information minimization. **MoCME** (Li, 2025) incorporates complementarity-aware fusion of modality experts, and **HERGC** (Xiao & Zhang, 2025) retrieves heterogeneous expert representations and leverages a generative decoder for link prediction. Despite their success,

---

[1] The source code detail is provided in Appendix D.1.

gating strategies often lead to load imbalance, *i.e.*, overusing few experts, resulting in expert collapse and inefficient training (Kool et al., 2021; Clark et al., 2022; Liu et al., 2023b).

Consequently, the concept of **Optimal Transport (OT)** is introduced to reformulate MoE routing as a transportation problem, aligning inputs with experts in a balanced and interpretable manner (Zhang et al., 2023). Formally, considering a transport scenario between suppliers $\mathbf{U} = \{u_i \mid i = 1, \ldots, |\mathbf{U}|\}$ and customers $\mathbf{V} = \{v_j \mid j = 1, \ldots, |\mathbf{V}|\}$, where $u_i$ and $v_j$ denote supply and demand masses respectively. Let $p_{ij} \geq 0$ be the quantity transported from $u_i$ to $v_j$ with cost $c_{ij} \geq 0$. The OT goal is to find an optimal transport plan $\mathbf{P}^* = \{p_{ij}^*\}$ that minimizes the total cost:

$$\mathbf{P}^* = \operatorname*{argmin}_{\mathbf{P}} \sum_{i=1}^{|\mathbf{U}|} \sum_{j=1}^{|\mathbf{V}|} p_{ij} c_{ij}, \quad \text{s.t.} \sum_{j=1}^{|\mathbf{V}|} p_{ij} = u_i, \ \sum_{i=1}^{|\mathbf{U}|} p_{ij} = v_j. \tag{1}$$

Several OT-based MoE routing strategies have been adopted. For example, **Sinkhorn-MoE** (Clark et al., 2022) reformulates expert routing as an entropy-regularized problem to ensure balanced expert assignment. **SVMoE** (Vesaghati & Zareapoor, 2024) incorporates expert capacity constraints and employs top-$k$ gating to promote both routing sparsity and load balance. Similarly, **FedOTP** (Liu et al., 2024) applies sparsity-constrained OT for expert routing in federated learning settings. However, existing OT-based MoE gating methods face several limitations, including *task-agnostic cost modeling, relation-unaware routing, and predefined distance metrics and expert prototypes*. To bridge these gaps, we propose integrating relational semantics into OT-based routing and dynamically constructing expert representations with a learnable cost function.

## 3 PROBLEM FORMULATION

A MultiModal Knowledge Graph (MMKG) is defined as $\mathcal{KG} = (\mathcal{E}, \mathcal{R}, \mathcal{T})$, where $\mathcal{E}$ denotes the set of entities, $\mathcal{R}$ the set of relations, and $\mathcal{T} = \{(h, r, t) \mid h, t \in \mathcal{E}, \ r \in \mathcal{R}\}$ the collection of relational triples. In addition to structural information, MMKGs also incorporate a modality set $\mathcal{M}$, where each modality provides complementary entity content, such as images (visual), textual descriptions (language), and *etc*. The goal of MMKG completion is to learn a scoring function $s(h, r, t) : \mathcal{E} \times \mathcal{R} \times \mathcal{E} \to \mathbb{R}^+$ that estimates the plausibility of a given triple $(h, r, t)$, where higher scores indicate greater likelihood of correctness. Typically, individual modality content is first encoded into latent representations, which are then fused into a unified entity embedding used by the scoring function to evaluate triple plausibility.

## 4 PROPOSED METHOD

The proposed method comprises two core modules: Modality-Specific Expert Group (MoSEG) and Relation-Guided Optimal Transport Routing with learnable cost (ReGOR) (see Fig. 1).

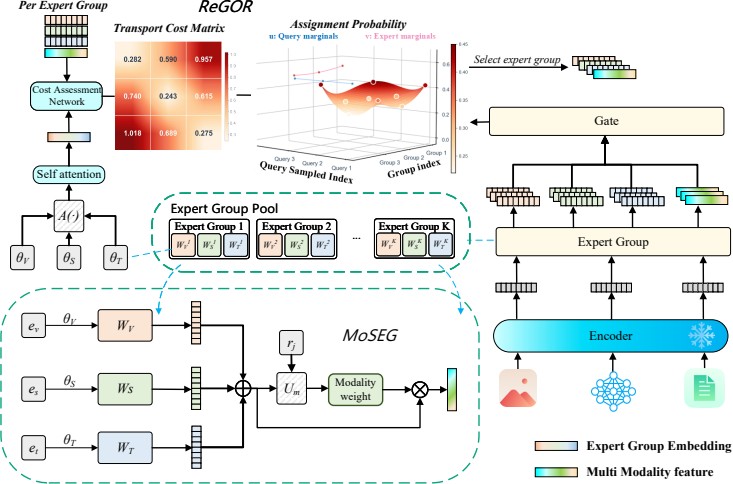

Figure 1: Overview of the proposed *ROAM* algorithm. MoSEG encodes inputs via modality-specific expert groups, while ReGOR performs relation-aware group selection using OT-based routing with learnable costs.

## 4.1 MODALITY-SPECIFIC EXPERT GROUP (MOSEG)

MoSEG is designed to promote effective specialization by decoupling expert responsibilities across modalities. As such, instead of sharing experts across multiple modalities, we assign each expert to one single modality. This modality-specific design aims to reduce cross-modality interference and enhances robustness by isolating modality-dependent variations.

Specifically, the proposed architecture consists of $K$ (a hyperparameter) expert groups, each comprising $|\mathcal{M}|$ sub-experts corresponding to the modalities in $\mathcal{M}$. For the $i$-th entity $e_i \in \mathcal{E}$ and its $m$-th modality $m \in [1, |\mathcal{M}|]$, a raw modality-specific feature vector $\boldsymbol{e}_{im}$ is first extracted. For instance, pre-trained models such as VGG (Simonyan & Zisserman, 2015) and BERT (Devlin et al., 2019) can be employed for the visual and textual modalities, respectively, to obtain initial representations. In contrast, structural and relational features are randomly initialized and subsequently optimized during training. Each extracted feature $\boldsymbol{e}_{im}$, together with the $j$-th relation feature $\boldsymbol{r}_j$, is fed into all $K$ expert groups in parallel.

Within the $k$-th expert group ($\mathcal{W}^{(k)}$, $k \in [1, K]$), sub-expert $\mathcal{W}_m^{(k)}$, corresponding to the $m$-th modality, is instantiated as a multilayer perceptron (MLP) with a single hidden layer and transforms the initialized modality-specific feature vector $\boldsymbol{e}_{im}$ into a modality-specific embedding:

$$\boldsymbol{v}_{im}^{(k)} = \mathcal{W}_m^{(k)}(\boldsymbol{e}_{im}; \boldsymbol{\theta}_m^{(k)}), \tag{2}$$

where $\boldsymbol{\theta}_m^{(k)}$ denotes the learnable parameters of expert $\mathcal{W}_m^{(k)}$. The outputs $\{\boldsymbol{v}_{im}^{(k)}\}_{m=1}^{|\mathcal{M}|}$ from modality-specific experts in group $k$ are aggregated **under a relation-aware context** to form the group-level entity representation for $e_i$. Specifically, a relation-aware attention mechanism is employed to compute the fused representation $\tilde{\boldsymbol{e}}_i^{(k)}$ within $\mathcal{W}^{(k)}$, based on the $j$-th relation embedding $\boldsymbol{r}_j$:

$$\tilde{\boldsymbol{e}}_i^{(k)} = \mathcal{W}^{(k)}(\boldsymbol{e}_i, \boldsymbol{r}_j) = \sum_{m=1}^{|\mathcal{M}|} w_m(\boldsymbol{v}_{im}^{(k)}, \boldsymbol{r}_j) \cdot \boldsymbol{v}_{im}^{(k)}, \tag{3}$$

where the attention weight is computed as:

$$w_m(\boldsymbol{v}_{im}^{(k)}, \boldsymbol{r}_j) = \mathcal{S}\left(\exp\left(\frac{\mathcal{U}_m\left([\boldsymbol{v}_{im}^{(k)} \parallel \boldsymbol{r}_j]\right) + \delta_m}{\tau}\right)\right), \tag{4}$$

where $\mathcal{S}$ denotes the softmax function, $\parallel$ represents vector concatenation, $\mathcal{U}_m$ is a learnable projection layer, $\delta_m \sim \mathcal{N}(0, \mathcal{U}_m([\boldsymbol{v}_{im}^{(k)} \parallel \boldsymbol{r}_j]))$ is learnable Gaussian noise (Cheng et al., 2020), and $\tau$ is a temperature hyperparameter.

## 4.2 RELATION-GUIDED OT ROUTING (REGOR)

While the MoSEG module produces relation-aware entity representations $\tilde{\boldsymbol{e}}_i^{(k)}$ from each expert group, a crucial step remains: determining which expert group(s) to select for downstream reasoning. Previous studies have applied **Optimal Transport (OT)** for expert gating by constructing a cost matrix between inputs and experts (Clark et al., 2022). In such context, the cost $c_{ik}$ is typically defined as the similarity distance (such as cosine) between the input representation (*i.e.*, $\boldsymbol{e}_i$) and a prototype vector $\boldsymbol{g}_k$ of the $k$-th expert (with random initial values):

$$c_{ik} = \text{dist}(\boldsymbol{e}_i, \boldsymbol{g}_k). \tag{5}$$

This traditional strategy can be directly extended to our setting by treating each expert group as a unit expert and estimating the cost between the input entity and each group's prototype vector. While straightforward, this design presents several drawbacks. *First*, the cost function is task-agnostic, relying solely on input–expert group similarity distance without considering downstream objectives. *Second*, the dispatching mechanism is relation-unaware, assigning experts uniformly across contexts without considering relational semantics. *Third*, each expert group is represented by a prototype vector $\boldsymbol{g}_k$, which fails to reflect the dynamic behavior of its constituent experts.

To address these limitations, we propose **Relation-Guided OT Routing with learnable cost (Re-GOR)**, which explicitly aligns entities with expert groups by incorporating relational context into the transport cost. Formally, the cost between the $i$-th entity ($\boldsymbol{e}_i$) and the $k$-th expert group ($\mathcal{W}^{(k)}$), conditioned on relation $\boldsymbol{r}_j$, is defined as:

$$c_{ik} \mid \boldsymbol{r}_j = \phi\left(f(\boldsymbol{e}_i, \boldsymbol{r}_j) \| g(\mathcal{W}^{(k)})\right), \tag{6}$$

where $\|$ is the vector concatenation, $\phi(\cdot)$ is a relation-aware distance function with output in the non-negative real space $\mathbb{R}^+$. The function $f(\cdot)$ transforms the input entity representation into a task-adaptive space, while $g(\cdot)$ maps the functional form of the $k$-th expert group $\mathcal{W}^{(k)}$ to a vector representation. Notably, $\mathcal{W}^{(k)}$ refers to the *function form* of the $k$-th expert group, rather than the *function value* evaluated from input.

To begin with, $f(\boldsymbol{e}_i, \boldsymbol{r}_j)$ is defined as $f(\boldsymbol{e}_i) = \tilde{\boldsymbol{e}}_i^{(k)}$ (as shown in Eq. (3)). This formulation enables relation-aware representations and implicitly aligns with downstream objectives. Additionally, for $g(\mathcal{W}^{(k)})$, we represent each group as a parametric neural network with a fixed structure. That is, when the architecture is predefined, the function $\mathcal{W}^{(k)}$ can be fully specified by its learnable parameters. Therefore, we concretize the $k$-th expert group via the set of its parameters $\{\boldsymbol{\theta}_m^{(k)}\}_{m=1}^{|\mathcal{M}|}$. Accordingly, $g(\mathcal{W}^{(k)})$ is computed as:

$$g(\mathcal{W}^{(k)}) = \sum_{m=1}^{|\mathcal{M}|} \alpha_m^{(k)} \boldsymbol{\theta}_m^{(k)}, \tag{7}$$

where $\alpha_m^{(k)} = \mathcal{S}\left(\mathcal{A}(\boldsymbol{\theta}_m^{(k)})\right)$ denotes attention weights obtained from a function $\mathcal{A}(\cdot)$ (implemented as a single hidden layer MLP). This construction also ensures dimensional compatibility for $\phi(\cdot, \cdot)$ when instantiated with vector-based distance metrics, such as $\ell_2$-norm. Consequently, the proposed formulation enables dynamic construction of expert group representations that reflect the actual composition and specialization of each expert group. Unlike traditional task-agnostic prototypes, $g(\mathcal{W}^{(k)})$ captures adaptive processing behaviors, thereby enhancing the flexibility and expressiveness of the gating mechanism.

Finally, we instantiate $\phi(\cdot)$ as a learnable function rather than a fixed similarity metric (*e.g.*, cosine similarity). Specifically, we implement $\phi$ as an MLP mapping to $\mathbb{R}^+$, thereby offering greater modeling capacity and flexibility. Overall, to compute the conditional cost in Eq. (6), we transform the entity representation using $\mathcal{W}^{(k)}(\boldsymbol{e}_i, \boldsymbol{r}_j)$ (or $\tilde{\boldsymbol{e}}_i^{(k)}$ from Eq. (3)), and concatenate it with the expert group representation $g(\mathcal{W}^{(k)})$, resulting in the final input to $\phi$ as:

$$c_{ik} \mid \boldsymbol{r}_j = \phi\left(\mathcal{W}^{(k)}(\boldsymbol{e}_i, \boldsymbol{r}_j) \| \sum_{m=1}^{|\mathcal{M}|} \alpha_m^{(k)} \cdot \boldsymbol{\theta}_m^{(k)}\right).$$

We need to point out that, from existing (Table 1) where OT is employed as router, ReGOR is a conditional OT resembling the form of functional OT in (Zhu et al., 2024) where both sides of the transportation are functions, and it can also be regarded as a Kantorovich relaxation of the Monge problem (Perrot et al., 2016). We can rewrite Eq. (6) to a more abstract form as follows

$$c_{ik} \mid \boldsymbol{r}_j = \phi\left(\boldsymbol{e}_i \| \boldsymbol{r}_j \| \mathcal{W}^{(k)}\right). \tag{8}$$

That is, it stands for the transportation from input entities ($\boldsymbol{e}_i$) to expert group functions ($\mathcal{W}^{(k)}$) under the relation context ($\boldsymbol{r}_j$). Then Eq. (8) is an asymmetric functional OT. Adding $f(\cdot)$ to the formulation pushes forward the probability measure from the entity space to the space of expert functions, thereby transforming the task into a Monge problem. In this setting, the optimal transport map is parameterized by $f(\cdot)$, which belongs to a family of general computation graphs (as shown in Eq. (3)). At last, $g(\cdot)$ makes the cost computation feasible which can be seen as a mapping from expert function space to a vector space.

### 4.3 OVERALL OBJECTIVE

The training objective jointly optimizes triple plausibility and expert routing by integrating a link prediction loss with a transport-based regularization. The overall loss is formulated as:

$$\mathcal{L}_{\text{total}} = \mathcal{L}_{\text{triple}} + \lambda \cdot \mathcal{L}_{\text{OT}}, \tag{9}$$

where $\lambda$ is the penalty hyperparameter. The $\mathcal{L}_{\text{triple}}$ loss is associated with with training triples:

$$\mathcal{L}_{\text{triple}} = \sum_{(h,r,t)\in\mathcal{T}} \log \sigma(s(\tilde{\boldsymbol{e}}_h^{(k)}, r, \tilde{\boldsymbol{e}}_t^{(k)})) + \sum_{(h',r,t')\in\mathcal{T}^-} \log\left(1 - \sigma(s(\tilde{\boldsymbol{e}}_{h'}^{(k)}, r, \tilde{\boldsymbol{e}}_{t'}^{(k)}))\right), \tag{10}$$

where $\sigma(\cdot)$ is the sigmoid function, $\tilde{e}^{(k)}$ is the fused entity embedding from the chosen group $k$ (see Eq. (3)), $\mathcal{T}$ and $\mathcal{T}^-$ is the set of positive/negative triples, and $s(\cdot)$ are the scoring function (such as Tucker (Balazevic et al., 2019)). Additionally, $\mathcal{L}_{\text{OT}}$ represents the expected transport cost under the learned OT routing plan:

$$\mathcal{L}_{\text{OT}} = \sum_{i,j \in \mathcal{B}} \sum_{k=1}^{K} p_{ik} \cdot \left( c_{ik} \mid \boldsymbol{r}_j \right) + \tau \sum_{i \in \mathcal{B}} \sum_{k=1}^{K} p_{ik} \log p_{ik} \tag{11}$$

where $|\mathcal{B}|$ is the batch size, $p_{ik}$ denotes the assignment plan from entity $e_i$ to expert group $k$ (computed using the Sinkhorn algorithm (Cuturi, 2013)), and $\tau$ is the regularization term. This entropy-based regularization is introduced to not only retain the convexity of the transport problem but also to enforce strict convexity, thereby guaranteeing convergence of the Sinkhorn iterations (Cuturi, 2013).

## 5 EXPERIMENTS

### 5.1 EXPERIMENTAL SETUP

**Dataset.** We evaluate *ROAM* on four public MMKGC benchmarks: MKG-W, MKG-Y (Xu et al., 2022), DB15K (Liu et al., 2019), and KVC16K (Zhang et al., 2024b), using an 8:1:1 train/validation/test split. Detailed dataset statistics (such as entity/relation counts, split sizes, and image/text feature dimensions) are provided in Appendix Table 6.

**Baselines.** The proposed method is compared with four categories of MMKGC approaches. **(A) OT-based: OTKGE** (Cao et al., 2022) aligns modality distributions via Wasserstein distance to improve multimodal fusion; **OT-MEL** (Zhang et al., 2024d) introduces optimal-transport-guided correlation assignment, using OT to align multimodal and unimodal tokens between mentions and entities, supplemented with knowledge distillation to transfer OT plans into attention maps; **(B) MoE-based: MoSE** (Zhao et al., 2022) employs modality-specific experts with dynamic ensembling. **HERGC** (Xiao & Zhang, 2025) retrieves heterogeneous expert signals and decodes triples generatively. **MoCME** (Li, 2025) uses complementarity-aware fusion and entropy-guided negative sampling. **MoMoK** (Zhang et al., 2025) adopts relation-guided MoE with expert selection via mutual information. **(C) OT+MoE-based: Sinkhorn-MoE** (Clark et al., 2022) formulates routing as entropy-regularized OT. **(D) Other fusion: MMRNS** (Xu et al., 2022) enhances negative sampling using multi-modal descriptions. **MMKRL** (Lu et al., 2022) aligns multi-source knowledge with component alignment and adversarial training. **IMF** (Li et al., 2023) uses bilinear pooling and weighted fusion. **QEB** (Wang et al., 2023) adopts query-enhanced modality fusion. **VISTA** (Lee et al., 2023) aligns image-text pairs using transformer encoders. **AdaMF** (Zhang et al., 2024c) applies adaptive fusion with adversarial training. *Their results are either directly sourced from their original papers or obtained using their released source codes with default configurations.*

**Implementation Details.** All components of *ROAM* are implemented with simple MLP layers and trained using Adam with standard settings. Details about key hyperparameters (including expert groups, Sinkhorn steps, learning rates, and regularization) are deferred to Appendix D.2.

### 5.2 OVERALL PERFORMANCE

Table 2 presents the performance comparison of *ROAM* against 13 baselines across four MMKGC benchmarks, with all results averaged over five independent runs. *ROAM* consistently outperforms all competitors on three datasets and achieves the second-best performance on the remaining one. The effectiveness of *ROAM* is particularly evident in the Hits@1 metric, as it achieves relative improvements of 9.76% on DB15K and 9.62% on KVC16K over the second-best models. In particular, *ROAM* outperforms MoE-based approaches such as Sinkhorn-MoE, HERGC and MoMoK. While these methods adopt modular expert architectures, HERGC relies on retrieval-based fusion without dynamic expert selection. Sinkhorn-MoE utilizes the entropy regularization while MoMoK employs mutual information minimization. All of them lack fine-grained control under relational context, which lead to suboptimal routing decisions. In contrast, *ROAM* incorporates a relation-conditioned method with a learnable cost function and dynamically constructed group representations, enabling precise, context-sensitive expert routing aligned with task objectives.

Table 2: Comparison with baselines on four MMKGC datasets (Metric: MRR/Hit@K). Best results are in **bold**, second best are underlined. Methods with **\*** are not originally proposed for MMKGC, but are reproduced in our study to compare expert routing or OT strategies. "*Improv.*" is calculated against existing best method.

| Model | MKG-W | | MKG-Y | | DB15K | | | | KVC16K | | | |
|---|---|---|---|---|---|---|---|---|---|---|---|---|
| | MRR | Hit@1 | MRR | Hit@1 | MRR | Hit@1 | Hit@3 | Hit@10 | MRR | Hit@1 | Hit@3 | Hit@10 |
| *OT-based Methods* | | | | | | | | | | | | |
| OTKGE (Cao et al., 2022) | 34.36 | 28.85 | 35.51 | 31.97 | 23.86 | 18.45 | 25.89 | 34.23 | 8.77 | 5.01 | 9.31 | 15.55 |
| *OT-MEL (Zhang et al., 2024d) | 34.48 | 28.75 | 34.14 | 30.97 | 38.45 | 30.87 | 40.47 | 50.73 | 16.51 | 10.46 | 17.62 | 28.89 |
| *MoE-based Methods* | | | | | | | | | | | | |
| MoSE (Zhao et al., 2022) | 33.34 | 27.78 | 36.28 | 33.64 | 28.38 | 21.56 | 30.91 | 41.67 | 8.81 | 4.75 | 9.46 | 16.40 |
| HERGC (Xiao & Zhang, 2025) | 38.89 | 33.12 | 39.82 | 36.73 | 40.76 | 33.09 | 45.07 | 54.88 | 17.12 | 10.71 | 18.34 | 29.35 |
| MoCME (Li, 2025) | 37.79 | 30.81 | 40.37 | 36.21 | 39.62 | 29.71 | 42.15 | 55.36 | 18.32 | 11.23 | 19.45 | 30.76 |
| MoMoK (Zhang et al., 2025) | 35.89 | 30.38 | 37.91 | 35.09 | 39.57 | 32.38 | 43.45 | 54.14 | 16.87 | 10.53 | 18.26 | 29.20 |
| *OT+MoE-based Methods* | | | | | | | | | | | | |
| *Sinkhorn-MoE (Clark et al., 2022) | 35.68 | 29.77 | 35.35 | 33.58 | 38.89 | 31.24 | 42.88 | 51.32 | 17.90 | 10.83 | 18.04 | 29.56 |
| *Other Methods* | | | | | | | | | | | | |
| MMRNS (Xu et al., 2022) | 35.03 | 28.59 | 35.93 | 30.53 | 32.68 | 23.01 | 37.86 | 51.01 | 13.31 | 7.51 | 14.19 | 24.68 |
| MMKRL (Lu et al., 2022) | 30.10 | 22.16 | 36.81 | 31.66 | 26.81 | 13.85 | 35.07 | 49.39 | 8.78 | 3.89 | 8.99 | 18.34 |
| IMF (Li et al., 2023) | 34.50 | 28.77 | 35.79 | 32.95 | 32.25 | 24.20 | 36.00 | 48.19 | 12.01 | 7.42 | 12.82 | 21.01 |
| QEB (Wang et al., 2023) | 32.38 | 25.47 | 34.37 | 29.49 | 28.18 | 14.82 | 36.67 | 51.55 | 12.06 | 5.57 | 13.03 | 25.01 |
| VISTA (Lee et al., 2023) | 32.91 | 26.12 | 30.45 | 24.87 | 30.42 | 22.49 | 33.56 | 45.94 | 11.89 | 6.97 | 12.66 | 21.27 |
| AdaMF (Zhang et al., 2024c) | 34.27 | 27.21 | 38.06 | 33.49 | 32.51 | 21.31 | 39.67 | 51.68 | 15.26 | 8.56 | 16.71 | 28.29 |
| *ROAM* | **40.09** | **34.22** | 40.17 | 37.31 | **44.19** | **36.32** | **48.18** | **58.99** | **19.89** | **12.31** | **20.98** | **33.21** |
| ±std | (±1.22) | (±1.57) | (±1.03) | (±1.33) | (±1.12) | (±0.91) | (±1.44) | (±1.23) | (±1.18) | (±1.24) | (±1.05) | (±1.14) |
| *Improv.* | +3.09% | +3.32% | -0.50% | +1.58% | +8.42% | +9.76% | +6.90% | +6.56% | +8.57% | +9.62% | +7.86% | +7.97% |

## 5.3 Ablation Studies

**Effect of MoSEG.** To validate our MoSEG design, we compare it with two variants. All of them share the same OT-based routing module (ReGOR) for fair comparison. Additionally, in all cases, each expert is implemented as a MLP with a single hidden layer, and takes the concatenated entity and relation embeddings as input.

- **Shared Expert Pool (SEP)** uses a single shared expert pool (with $3 \times K$ experts) across all three modalities, without modality-specific separation.
- **Modality-Gated Experts (MGE)** employs separate expert pools (each with $K$ experts) for structural, image, and text modalities.

Results from Table 3 validate our architectural design. The SEP variant, which processes all modalities within a single expert pool, yields the lowest performance. This indicates that forcing experts to handle heterogeneous inputs without dedicated separation leads to functional interference. The MGE model, by employing separate expert pools for each modality, outperforms SEP, highlighting the benefit of modality-specialized processing. However, its performance is still surpassed by MoSEG, as MGE performs routing independently for each modality, preventing it from capturing cross-modal dependencies. In contrast, the proposed MoSEG architecture achieves the best results by organizing modality-specific sub-experts into unified expert groups, enabling more effective cross-modal routing and integration.

Table 3: Impact from the MoSEG architecture across four benchmarks. Metric: MRR (%).

| Variants | MKG-W | MKG-Y | DB15K | KVC16K |
|---|---|---|---|---|
| SEP | $38.87 \pm 1.35$ | $39.21 \pm 1.28$ | $42.89 \pm 1.51$ | $18.75 \pm 1.12$ |
| MGE | $39.45 \pm 1.11$ | $39.78 \pm 1.05$ | $43.52 \pm 1.33$ | $19.31 \pm 0.98$ |
| **MoSEG** | $\mathbf{40.09 \pm 1.22}$ | $\mathbf{40.17 \pm 1.03}$ | $\mathbf{44.19 \pm 1.12}$ | $\mathbf{19.89 \pm 1.18}$ |

**Effect of ReGOR.** To evaluate the effectiveness of the proposed ReGOR routing, a comprehensive ablation study is conducted. Our full model employs a relation-aware cost function defined as $\phi\big(f(\boldsymbol{e}_i, \boldsymbol{r}_j) \| g(\mathcal{W}^{(k)})\big)$ (see Eq. (6)). To isolate the contribution of each component, we systematically consider the following model variants:

- **w/o** $f(\boldsymbol{e}_i, \boldsymbol{r}_j)$ removes entity transformation and relational condition, but concatenating $\boldsymbol{e}_i$ and $\boldsymbol{r}_j$ directly with retaining dynamic group representation: $\phi(\boldsymbol{e}_i \| \boldsymbol{r}_j \| g(\mathcal{W}^{(k)}))$.
- **w/o** $g(\mathcal{W}^{(k)})$ replaces dynamic group representation with prototype $v_k$, keeping other components: $\phi(f(\boldsymbol{e}_i, \boldsymbol{r}_j) \| \boldsymbol{v}_k)$.
- **w/o** $\phi(\cdot)$ employs predefined cost function: $1 - \cos(f(\boldsymbol{e}_i, \boldsymbol{r}_j), g(\mathcal{W}^{(k)}))$.

The ablation results in Table 4 highlight the contribution of each component. Removing the entity–relation transformation module $f(\boldsymbol{e}_i, \boldsymbol{r}_j)$ consistently degrades performance, as it prevents the

Table 4: Ablation study on the components of our routing architecture. Metric: MRR (%).

| Variants | MKG-W | MKG-Y | DB15K | KVC16K |
|---|---|---|---|---|
| w/o $f(\boldsymbol{e}_i, \boldsymbol{r}_j)$ | $39.98 \pm 0.94$ | $40.07 \pm 1.36$ | $43.38 \pm 0.78$ | $19.12 \pm 0.83$ |
| w/o $g(\mathcal{W}^{(k)})$ | $39.61 \pm 1.19$ | $39.66 \pm 1.53$ | $43.76 \pm 1.06$ | $19.41 \pm 0.89$ |
| w/o $\phi(\cdot)$ | $39.42 \pm 0.82$ | $39.50 \pm 1.17$ | $43.71 \pm 1.28$ | $19.37 \pm 0.96$ |
| Full | $40.09 \pm 1.22$ | $40.17 \pm 1.03$ | $44.19 \pm 1.12$ | $19.89 \pm 1.18$ |

model from generating relation-aware representations and capturing entity–relation interactions. Excluding the dynamic group representation $g(\mathcal{W}^{(k)})$ leads to noticeable drops, since expert prototypes lack the flexibility to adapt to task-specific semantics. Most significantly, disabling the learnable cost $\phi(\cdot)$ results in the largest performance loss, highlighting its importance in enabling adaptive expert selection through task-specific transport distances. Together, these findings demonstrate that the full model benefits from the complementary effects of all three components, *i.e.*, relation condition, dynamic expertise, and learnable cost-based routing, yielding optimal performance.

**Expert Group Number.** To investigate the effect of the number of expert groups $K$, we vary it from 3 to 7 and evaluate both model performance (MRR) and routing entropy on the DB15K and MKG-W datasets. Routing entropy, computed as the Shannon entropy over expert selection probabilities, measures the diversity of expert usage, where higher values indicate more uniform expert routing. On DB15K (Fig.2 (a)), performance improves as the number of expert groups increases, peaking at 5 groups (MRR = 44.19%), after which it declines. A similar trend is observed on MKG-W (Fig.2(b)), where the best MRR (40.09%) is achieved with 4 expert groups before performance drops. These results indicate that the optimal number of expert groups may vary across datasets depending on their size and complexity. Nevertheless, using a moderate number of

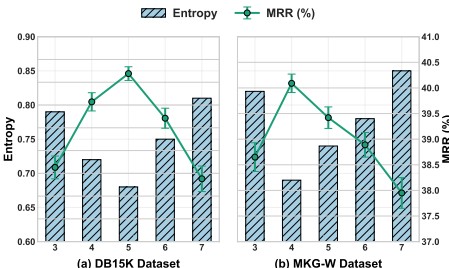

Figure 2: Effect of expert-group number on model performance (MRR) and routing entropy, where entropy measures the uncertainty of routing decisions.

groups generally yields stable and competitive performance across benchmarks. Additionally, routing entropy also reaches its minimum with these optimal configurations on both datasets, suggesting that improved performance correlates with more confident and decisive group selection.

**Effect of Modality Fusion.** To better understand the contribution of each modality to the overall reasoning capability, a series of ablation variants are constructed by enabling different combinations of modalities (structure, text, image), including single-modality and bi-modality configurations.

Table 5: Effect of different modality combinations with *ROAM*. Metric: MRR (%).

| Modality Combination | MKG-W | MKG-Y | DB15K | KVC16K |
|---|---|---|---|---|
| *Single Modality Baselines* | | | | |
| Structure only | $37.41 \pm 2.13$ | $37.37 \pm 2.05$ | $41.11 \pm 2.45$ | $17.89 \pm 1.04$ |
| Image only | $37.35 \pm 1.86$ | $37.29 \pm 2.21$ | $41.02 \pm 1.91$ | $17.82 \pm 1.33$ |
| Text only | $37.12 \pm 2.41$ | $37.03 \pm 1.89$ | $40.73 \pm 2.28$ | $17.58 \pm 0.87$ |
| *Two Modality Combinations* | | | | |
| w/o Structure (Img + Txt) | $39.35 \pm 1.95$ | $39.30 \pm 2.24$ | $43.23 \pm 1.82$ | $19.15 \pm 1.18$ |
| w/o Image (Struct + Txt) | $39.15 \pm 2.38$ | $39.11 \pm 1.88$ | $43.02 \pm 2.47$ | $18.98 \pm 1.03$ |
| w/o Text (Struct + Img) | $39.38 \pm 2.09$ | $39.31 \pm 2.16$ | $43.24 \pm 2.31$ | $19.17 \pm 1.27$ |
| **Full Model (all)** | $\mathbf{40.09 \pm 1.22}$ | $\mathbf{40.17 \pm 1.03}$ | $\mathbf{44.19 \pm 1.12}$ | $\mathbf{19.89 \pm 1.18}$ |

Table 5 reports the MRR results across four benchmark datasets under each modality combination. Each modality is found to independently contribute to the model's reasoning capability. For single modality, the structure-only case achieves the highest MRR, followed by the image-only model, while the text-only one performs the worst. This may be attributed to the relatively sparse or noisy nature of textual descriptions compared to structured triples and visual features.

In two-modality settings, the model consistently outperforms single-modality variants, highlighting the complementarity among modalities. Not surprisingly, the full (three)-modality model outperforms all single and partial combinations, demonstrating that the proposed *ROAM* effectively integrates heterogeneous modality information and enhances the representations of entities and relations for the downstream MMKGC task.

**Hyperparameter Analysis.** This section presents a sensitivity analysis by tuning one hyperparameter at a time while keeping all others fixed. The analysis focuses on three hyperparameters: the learning rate $\eta$, the Sinkhorn iteration $L$ (the OT-plan solver), and the OT regularization term $\lambda$ (from Eq. (9)), and their results are shown in Fig. 3.

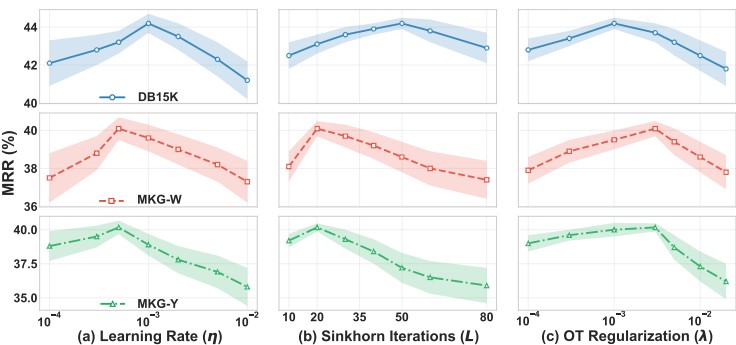

Figure 3: MRR Performance to key hyperparameters.

A consistent trend is observed across all benchmarks, that is, model performance improves with increasing hyperparameter values up to an optimal point, beyond which it declines. On DB15K, the best performance is achieved with $r_l = 0.001$, $L = 50$, and $\lambda = 0.001$, while on MKG-W and MKG-Y, optimal results are obtained with a lower $r_l$ (0.0005), fewer $L$ (20), and a higher $\lambda$ (0.005), reflecting differences in dataset scale and complexity. Notably, excessively large $r_l$ or $\lambda$ cause significant performance degradation, likely due to unstable optimization and over-constrained transport plans, respectively. In addition, both insufficient and excessive $L$ hinder group routing quality: too few iterations result in poorly approximated transport plans, while too many may lead to over-smoothing, diminishing the distinctiveness of group selection.

**Efficiency Analysis.** To evaluate computational efficiency, the model training time and GPU memory usage of *ROAM* are compared with the following baselines: MoMoK, MMKRL, OTKGE, and MMRNS. our method achieves a favorable trade-off between performance and efficiency. As of training time, *ROAM* requires 11.4 seconds, slightly longer than MMKRL (7.5s) and MoMoK (9.8s), but significantly faster than OTKGE (25.5s) and MMRNS (70.1s). This efficiency primarily stems from conditioning the routing process on relation representations. By incorporating relation-aware condition, the model narrows the selection space, enabling more

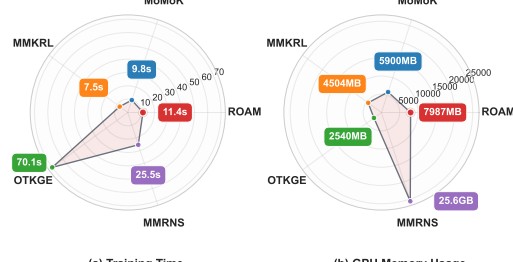

Figure 4: Comparison of model fine-tuning time and GPU memory usage across MMKGC methods.

targeted and decisive routing. In terms of memory usage, *ROAM* consumes 7987MB of GPU memory, moderately higher than MMKRL (4504MB), MoMoK (5900MB), and OTKGE (2540MB), but significantly lower than MMRNS (25.6GB). Compared to OTKGE, *ROAM* consumes more memory due to the inclusion of expert modules. Yet, unlike MMRNS, our method avoids maintaining densely activated experts and global fusion layers, significantly reducing memory overhead. Overall, the results confirm that *ROAM* achieves an effective balance between capacity and efficiency, while remaining computationally affordable.

## 6 CONCLUSION

This paper presents **ROAM**, a novel framework for Multimodal Knowledge Graph Completion that integrates Modality-Specific Expert Group encoding (MoSEG) with Relation-Guided Optimal Transport Routing (ReGOR). By incorporating relational semantics into a learnable OT-based gating mechanism and dynamically constructing group representations via network parameters, *ROAM* enables fine-grained, context-aware expert selection. Extensive experiments across multiple MMKGC benchmarks demonstrate that *ROAM* consistently outperforms existing MoE- and OT-based methods. Future work includes extending *ROAM* to exploring scalable expert adaptation, integrating additional modalities, and further reducing the computational overhead of Optimal Transport.

ETHICS STATEMENT

This work adheres to the ICLR Code of Ethics. The research did not involve any human subjects or animal experimentation. All datasets used in our experiments, including MKG-W, MKG-Y, DB15K, and KVC16K, are publicly available benchmark datasets and were employed in compliance with their respective usage guidelines. We have taken care to avoid introducing biases or discriminatory outcomes during data processing, model design, and evaluation. No personally identifiable information (PII) was used in this study, and no experiments were conducted that could raise privacy or security concerns. We are committed to maintaining transparency and integrity throughout the research process.

REPRODUCIBILITY STATEMENT

We have made substantial efforts to ensure the reproducibility of our work. The ROAM framework, including its implementation details, hyperparameter settings, and evaluation protocols, are comprehensively documented in the main paper and appendix. The source code has been made publicly available in an anonymous repository (see Appendix D.1), and we provide detailed descriptions of the datasets, preprocessing steps, and training configurations. Results reported in the paper are averaged over five independent runs to reduce randomness. We believe that these resources and explanations will enable the community to faithfully reproduce our results and further build upon them.

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

## A  DATASET STATISTICS

Table 6: Statistics of multimodal knowledge graph datasets.

| Dataset | #Entities | #Relations | Train | Valid | Test | Img/Text Dim |
|---|---|---|---|---|---|---|
| MKG-W | 15,000 | 169 | 34,196 | 4,276 | 2,665 | 256 / 256 |
| MKG-Y | 15,000 | 28 | 21,310 | 2,663 | 2,663 | 256 / 256 |
| DB15K | 12,842 | 279 | 79,222 | 9,902 | 9,904 | 4096 / 768 |
| KVC16K | 16,015 | 4 | 180,190 | 22,523 | 22,525 | 768 / 768 |

## B  CASE STUDY AND INTERPRETABILITY

To further assess the interpretability and dynamic routing behavior of our model, we conduct a case study from two complementary perspectives: (1) **context-aware expert group selection**, and (2) **intra-group modality preference**.

### B.1  CONTEXT-AWARE EXPERT GROUP SELECTION

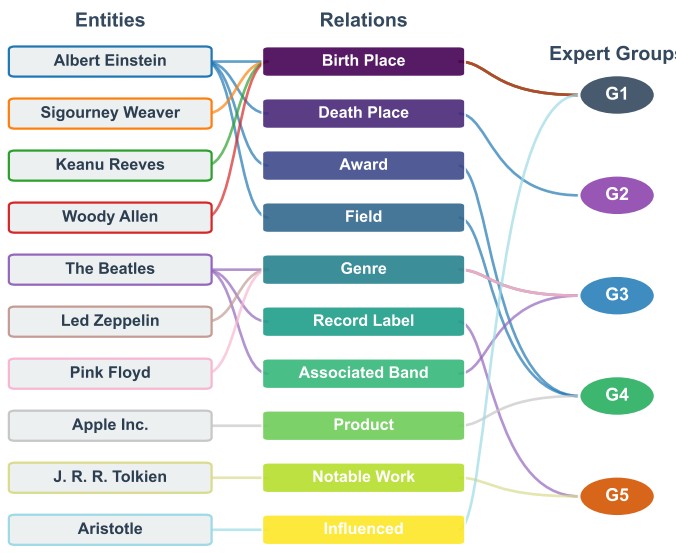

Figure 5: Representative routing results across relations and head entities in DB15K.

Fig. 5 illustrates representative examples from the DB15K dataset, highlighting how our model routes inputs to different expert groups based on the semantics of entities and relations. Notably, the model demonstrates efficient context-aware routing capabilities:

- **Semantic Clustering Routing:** The model learns to route queries (i.e., the head entity and relation) with similar semantics to specific expert groups. For instance, for the relation *Birth Place*, diverse entities such as *Albert Einstein*, *Sigourney Weaver*, and *Keanu Reeves* are all consistently routed to Group 1 (G1). Similarly, for the relation *Genre*, musical acts like *The Beatles*, *Led Zeppelin*, and *Pink Floyd* are all directed to Group 3 (G3).

- **Entity-Aware Context Switching:** Importantly, the model also showcases dynamic routing even with the *same head entity* but under different relations. For example, the head entity *Albert Einstein* is routed to three different groups: to Group 1 (G1) for the relation *Birth Place*, to Group 2 (G2) for *Death Place*, and to Group 4 (G4) for both *Award* and *Field*, respectively.

These routing patterns demonstrate that our model has successfully learned to associate specific expert groups with distinct semantic domains (e.g., G1 for geographical facts, G3 for music attributes), while also being able to dynamically select the most appropriate group conditioned on the fine-grained entity context under the context of input relations.

### B.2 COMPARISON OF EXPERT ROUTING FRAMEWORKS VIA MODALITY UTILIZATION

To visually demonstrate the superiority of our dynamic, relation-guided routing mechanism, we compare its effect on modality utilization against two representative Mixture-of-Experts (MoE) frameworks, that is:

- **MoMoK (MoE) (Zhang et al., 2025):** This method processes each modality in separate, expert-based manner. The final prediction is a weighted aggregation from each expert, governed by a score-level gating network.
- **Sinkhorn-MoE (OT+MoE) (Clark et al., 2022):** This method formulates the expert selection as an entropy-regularized Optimal Transport problem. The objective is to find a transport plan that minimizes assignment cost, to balance the computational load across experts.

Fig. 6 presents the modality preferences for the experts or groups in each framework, which reflects the efficacy of the employed routing strategy. All experiments are conducted on the DB15K dataset. As observed, **MoMoK** assigns a separate expert group to each modality and aggregates their outputs via a score-level gating mechanism. While this design ensures balanced utilization within each modality-specific path, the final gating often exhibits a bias toward one dominant modality (e.g., text), likely due to its consistently higher score outputs. This limits the model's ability to dynamically adjust expert activation based on input-specific factors such as relation type or modality relevance. The **Sinkhorn-MoE** suffers from severe modality collapse, where most inputs are routed to a few dominant experts, likely caused by the lack of relation-aware guidance and insufficient regularization during routing. In contrast, our **ROAM** approach produces diverse and well-specialized expert groups, each exhibiting clear and distinct modality preferences. This is attributed to the integration of relation-guided expert selection with the proposed OT, which jointly promote both specialization and diversity in expert activation.

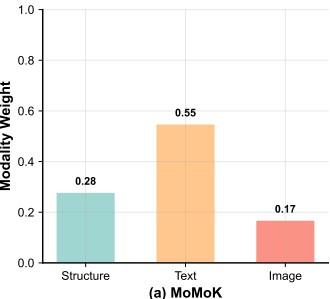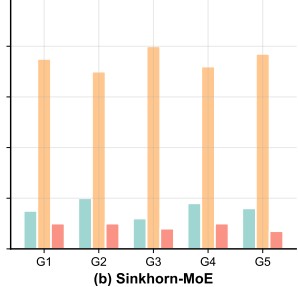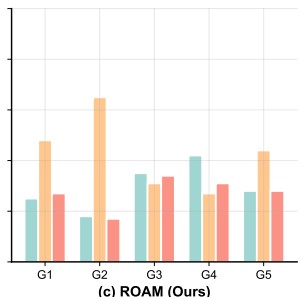

Figure 6: Average modality weight distributions resulting from different MoE based routing methods on the DB15K dataset.

### B.3 STATISTICAL SIGNIFICANCE TEST

To further confirm the effectiveness of our method, we conduct pairwise statistical significance tests using the Wilcoxon signed-rank test (Wilcoxon, 1945) on link prediction results. The test compares our model (ROAM) with strong baselines on standard evaluation metrics (MRR and Hits@1), based on per-relation performance distributions. We report the $p$-values for each comparison in Table 7. A threshold of $p < 0.005$ is used to determine statistical significance. Results show that our model significantly outperforms prior methods across most datasets and metrics.

Table 7: Wilcoxon signed-rank test results comparing ROAM against key MoE-based baselines. For each baseline, we report its performance, our model's performance, and the two-sided $p$-value. Bold indicates statistical significance ($p < 0.005$).

| Model | MKG-W | | MKG-Y | | DB15K | | KVC16K | |
|---|---|---|---|---|---|---|---|---|
| | MRR | Hit@1 | MRR | Hit@1 | MRR | Hit@1 | MRR | Hit@1 |
| MoSE (Zhao et al., 2022) | 33.34 | 27.78 | 36.28 | 33.64 | 28.38 | 21.56 | 8.81 | 4.75 |
| *ROAM* | 40.09 | 34.22 | 40.17 | 37.31 | 44.19 | 36.32 | 19.89 | 12.31 |
| *p-value* | *1.3e-5* | *2.1e-5* | *1.8e-4* | *2.5e-4* | *1.2e-6* | *9.0e-7* | *4.5e-7* | *5.1e-7* |
| HERGC (Xiao & Zhang, 2025) | 38.89 | 33.12 | 39.82 | 36.73 | 40.76 | 33.09 | 17.12 | 10.71 |
| *ROAM* | 40.09 | 34.22 | 40.17 | 37.31 | 44.19 | 36.32 | 19.89 | 12.31 |
| *p-value* | *2.5e-4* | *1.8e-4* | *8.5e-4* | *9.1e-4* | *1.2e-5* | *9.0e-6* | *4.0e-5* | *8.1e-5* |
| MoCME (Li, 2025) | 37.79 | 30.81 | 40.37 | 36.21 | 39.62 | 29.71 | 18.32 | 11.23 |
| *ROAM* | 40.09 | 34.22 | 40.17 | 37.31 | 44.19 | 36.32 | 19.89 | 12.31 |
| *p-value* | *3.1e-4* | *2.2e-4* | *7.8e-2* | *2.1e-4* | *1.5e-5* | *2.1e-5* | *3.3e-5* | *2.9e-5* |
| MoMoK (Zhang et al., 2025) | 35.89 | 30.38 | 37.91 | 35.09 | 39.57 | 32.38 | 16.87 | 10.53 |
| *ROAM* | 40.09 | 34.22 | 40.17 | 37.31 | 44.19 | 36.32 | 19.89 | 12.31 |
| *p-value* | *8.8e-5* | *6.5e-5* | *2.1e-4* | *1.4e-4* | *1.4e-5* | *8.8e-6* | *7.3e-6* | *8.0e-6* |

## C  ADDITIONAL EXPERIMENTS AND ANALYSIS

To further validate the scalability, component necessity, and efficiency of ROAM, we conduct a series of additional experiments. This section details the performance on large-scale knowledge graphs, a component-wise analysis, an evaluation of routing solvers, and an analysis of relational context.

### C.1  SCALABILITY ON LARGE-SCALE KNOWLEDGE GRAPHS

To assess the performance of *ROAM* on large-scale datasets, we conduct experiments on **FB15k-237-IMG** ($|\mathcal{E}| = 14,541, |\mathcal{R}| = 237, |\mathcal{T}| = 310,116$), which contains nearly $1.7\times$ the triples of KVC16K. Two recent baselines, including MEOW (Zhao et al., 2025) and DM-MKGC (Liu & Ren, 2025), are also included for comparison purposes.

Table 8: Performance comparison on the large-scale FB15k-237-IMG dataset. Note that * denotes results are directly sourced from original papers and - denotes results that are not reported in original.

| Model | MRR% | Hit@1 (%) | Hit@3 (%) | Training Time (s) |
|---|---|---|---|---|
| MoMoK | 35.52 | 26.07 | 39.32 | 26.9 |
| MMRNS | 33.16 | 25.02 | 35.80 | 192.7 |
| OTKGE | 32.81 | 24.26 | 35.67 | 72.5 |
| MMKRL | 31.45 | 23.05 | 34.26 | 23.0 |
| MEOW | 37.9* | 28.10* | 39.10* | 62.2* |
| DM-MKGC | 34.6* | 26.50* | 37.10* | - |
| *ROAM* | **42.66** | **33.39** | **46.77** | **32.1** |

Table 8 compares *ROAM* against key baselines. Despite the increase in graph size, *ROAM* consistently outperforms existing KGC baselines. In terms of training efficiency, *ROAM* is approximately $6\times$ faster than MMRNS and more than $2\times$ faster than OTKGE and MEOW, while remaining comparable to lightweight MoE-based methods such as MoMoK. These results demonstrate that *ROAM* scales effectively to large KGs while maintaining strong performance.

### C.2  COMPONENT NECESSITY

Table 4 evaluates the three sub-components of ReGOR within our full functional-OT formulation, as shown in Eq. (6). Specifically, each ablation removes only one component at a time, while the other

two still remain active. Results reveal that three sub-components of ReGOR are functionally complementary; when one component is removed, the system continues to benefit from the remaining two, which masks the individual degradation.

Yet, to more clearly isolate the contribution of each sub-component, we further conduct an experiment using a conventional OT–MoE routing cost as the **Baseline**:

$$c_{ik} \mid \boldsymbol{r}_j = 1 - \cosine(\boldsymbol{e}_i \| \boldsymbol{r}_j, \boldsymbol{v}_k), \tag{12}$$

where $\boldsymbol{v}_k$ denotes the $k$-th expert prototype. Starting from this baseline, accordingly, we reintroduce our three components one at a time. The following variants are constructed:

- **+LearnableCost:** Replaces the fixed cosine distance with the learnable cost network, formulated as $c_{ik} \mid \boldsymbol{r}_j = \phi(\boldsymbol{e}_i \| \boldsymbol{r}_j \| \boldsymbol{v}_k)$.
- **+DynGroupRep:** Replaces the prototypes $\boldsymbol{v}_k$ with our dynamic group representation derived from expert weights, formulated as $c_{ik} \mid \boldsymbol{r}_j = 1 - \cosine(\boldsymbol{e}_i \| \boldsymbol{r}_j, g(\mathcal{W}^{(k)}))$.
- **+RelTransform:** Incorporates the relation-conditioned entity transformation before computing distance, formulated as $c_{ik} \mid \boldsymbol{r}_j = 1 - \cosine(f(\boldsymbol{e}_i, \boldsymbol{r}_j), \boldsymbol{v}_k)$.

Table 9: Ablation study of ReGOR sub-components (Metric: MRR %).

| Model Variant | MKG-W | MKG-Y | DB15K | KVC16K |
|---|---|---|---|---|
| Baseline | 36.42 | 36.74 | 39.22 | 18.05 |
| +LearnableCost | 37.51 | 37.95 | 40.58 | 18.82 |
| +DynGroupRep | 37.78 | 38.34 | 40.15 | 18.95 |
| +RelTransform | 37.79 | 39.18 | 41.28 | 18.65 |

The results are reported in Table 9, where each component produces a significant and consistent performance improvement over the baseline. This confirms that the design elements of ReGOR are individually necessary to bridge the gap between standard OT approaches and our full model.

## C.3 EFFICIENCY OPTIMIZATION AND ROBUSTNESS ANALYSIS

**Efficiency of routing solvers.** To better understand the efficiency of ReGOR, we implement three variants of the routing solver, in addition to the traditional Sinkhorn algorithm:

- **Top-K Sparse Routing (TKSR):** Removes the iterative OT optimization and directly routes each token to the $k = 2$ lowest-cost expert groups based on the cost matrix (Shazeer et al., 2017).
- **Sparse Optimal Transport (SOT):** Solves the OT problem but enforces sparsity by masking out entries in the cost matrix whose values exceed a threshold $\tau = 0.1$ (Liu et al., 2023a).
- **Low-rank Approximate OT (LOT):** Employs a low-rank matrix factorization on the cost matrix and reduces the number of Sinkhorn iterations by 50% (i.e., $L = 25$) (Scetbon et al., 2021).

Table 10 summarizes the results across four benchmarks. Although these variants offer slight speed benefits, they consistently incur performance degradation. The limited speedup can be explained by the computational complexity. The total single-step complexity of ROAM is $\mathcal{O}(MKd^2 + LK^2)$. Since the embedding dimension ($d$) is typically much larger than the number of expert groups ($K$) and Sinkhorn iterations ($L$) (i.e., $d \gg K, L$), the **encoding term** $\mathcal{O}(MKd^2)$ dominates the overall cost. Consequently, methods like TKSR or LOT, which reduce the routing complexity $\mathcal{O}(LK^2)$, have a limited effect on the end-to-end runtime.

**Robustness to Noise.** To evaluate the robustness of *ROAM* under conditions of weak or noisy semantics, a perturbation experiment is performed via injecting random Gaussian noise (10% and 20%) into the input representations. As shown in Table 11, *ROAM* exhibits minimal performance degradation even under high noise levels, demonstrating the stability of the proposed relation-aware routing method.

Table 10: Efficiency and performance comparison of routing strategies (Metric: MRR / Hit@1 in %).

| Dataset | Method | MRR | Hit@1 | Training (s) | Inference (s) |
|---------|--------|-----|-------|--------------|---------------|
| **MKG-W** | TKSR | 39.18 | 32.86 | 4.87 | 6.47 |
| | SOT | 39.47 | 33.15 | 4.88 | 6.49 |
| | LOT | 39.80 | 33.94 | 4.89 | 6.51 |
| | **ROAM** | **40.09** | **34.22** | **4.93** | **6.53** |
| **MKG-Y** | TKSR | 39.75 | 36.66 | 3.11 | 6.48 |
| | SOT | 39.84 | 36.81 | 3.12 | 6.50 |
| | LOT | 39.94 | 37.09 | 3.14 | 6.52 |
| | **ROAM** | **40.17** | **37.31** | **3.16** | **6.56** |
| **DB15K** | TKSR | 43.03 | 34.41 | 11.35 | 20.48 |
| | SOT | 43.90 | 35.87 | 11.36 | 20.49 |
| | LOT | 44.14 | 36.09 | 11.38 | 20.51 |
| | **ROAM** | **44.19** | **36.32** | **11.42** | **20.54** |
| **KVC16K** | TKSR | 19.25 | 11.82 | 25.65 | 27.15 |
| | SOT | 19.72 | 12.15 | 25.72 | 27.22 |
| | LOT | 19.85 | 12.28 | 25.78 | 27.28 |
| | **ROAM** | **19.89** | **12.31** | **25.82** | **27.32** |

Table 11: Robustness analysis against noise (Metric: MRR / Hit@1 in %).

| Dataset | Noise-free (0%) | 10% Noise | 20% Noise |
|---------|-----------------|-----------|-----------|
| **MKG-W** | 40.09 / 34.22 | 39.33 / 33.41 | 38.94 / 33.07 |
| **MKG-Y** | 40.17 / 37.31 | 39.75 / 36.98 | 39.56 / 36.56 |
| **DB15K** | 44.19 / 36.32 | 42.51 / 34.42 | 41.85 / 33.06 |
| **KVC16K** | 19.89 / 12.31 | 19.23 / 11.85 | 18.97 / 11.57 |

## C.4 ROLE OF RELATIONAL CONTEXT IN ROUTING

The relational information ($r$) is employed in both the expert encoding and routing phases. It is worth pointing out that the reuse of $r$ is not a redundant design choice. Instead, it reflects two functionally distinct roles crucial for reasoning:

- **Modulation Role (Intra-group):** Inside each expert group, $r$ modulates the fusion of structural, textual, and visual sub-experts (Eqs. (3)–(4)). The goal is to obtain a relation-aware group representation.

- **Decision Role (Routing):** In the ReGOR routing, $r$ defines the conditional OT transport cost $c_{ik} \mid r = \phi(f(e_i, r) \| g(\mathcal{W}^{(k)}))$ (Eq. (6)). This determines which expert group is selected based on semantic compatibility.

Theoretically, forcing the routing module to ignore $r$ is equivalent to pushing all relation-dependent weights in the cost network toward zero. This yields a *degenerate solution* where the OT transport plan becomes relation-agnostic, creating a performance lower bound where no relational context contributes to group selection.

To empirically verify this, we evaluate a variant where $r$ is explicitly removed from the routing module. As shown in Table 12, removing the relational context leads to a consistent performance drop across all datasets. This confirms that using $r$ to define the conditional transport cost is essential for guiding the selection of appropriate expert groups, preventing the model from reverting to a relation-agnostic routing scheme.

Table 12: Ablation analysis of relation-based routing (Metric: MRR in %).

| Variant | MKG-W | MKG-Y | DB15K | KVC16K |
|---|---|---|---|---|
| with $r$ (Full) | **40.09** | **40.17** | **44.19** | **19.89** |
| w/o $r$ | 39.33 | 39.68 | 43.77 | 19.26 |

## D  IMPLEMENTATION DETAILS

### D.1  CODE REPOSITORY

To promote reproducibility and facilitate peer verification, we have released the full implementation of our framework in an anonymous repository at `https://anonymous.4open.science/ r/ROAM-8913`. This repository contains all necessary components to reproduce the main results presented in the paper.

### D.2  HYPERPARAMETER SETTINGS

Key hyperparameters and architectural choices for our model (ROAM) are detailed in Table 13. Common settings include using single-layer MLPs with ReLU activation for all core components (sub-experts, cost functions, gating projections), training with the Adam optimizer on a single NVIDIA RTX 4090 GPU, and using the Tucker scoring function for link prediction.

Table 13: Hyperparameter settings for ROAM.

| Hyperparameter | MKG-W | MKG-Y | DB15K | KVC16K |
|---|---|---|---|---|
| *Training Parameters* | | | | |
| Batch Size | 1024 | 1024 | 1024 | 1024 |
| Training Epochs | 2000 | 2000 | 2000 | 2000 |
| Learning Rate ($\eta$) | $5e^{-4}$ | $5e^{-4}$ | $1e^{-3}$ | $1e^{-3}$ |
| *Model Architecture Parameters* | | | | |
| Embedding Dimension | 256 | 256 | 256 | 256 |
| Task Embedding Dimension | 256 | 256 | 256 | 256 |
| Gating Hidden Dimension | 256 | 256 | 256 | 256 |
| Number of Expert Groups ($K$) | 4 | 4 | 5 | 5 |
| *Routing Parameters* | | | | |
| Sinkhorn Iterations ($L$) | 20 | 20 | 50 | 50 |
| OT Regularization ($\lambda$) | 0.005 | 0.005 | 0.001 | 0.001 |

## E  COMPUTATIONAL COMPLEXITY

To analyze the computational cost of our proposed model (ROAM), we evaluate its time complexity for a single training step (per batch). Let $K$ be the number of expert groups, $M$ be the number of modalities, $d$ be the embedding dimension, and $L$ be the number of Sinkhorn iterations for the Optimal Transport solver. The total complexity can be decomposed into two main parts:

- **MoSEG (Modality-Specific Expert Group) Encoding:** Each of the $M$ modality features (dimension $d$) is processed by $K$ expert groups. Within each group, $M$ sub-expert MLPs transform the features, leading to a complexity of $O(MKd^2)$.

- **ReGOR (Relation-Guided Optimal Transport Routing):** The gating network computes a relation-specific cost matrix of size $K \times K$ to assign the relation query to the expert groups. This is followed by $L$ iterations of the Sinkhorn algorithm, resulting in a complexity of $O(LK^2)$.

Consequently, the overall single-step time complexity of *ROAM* is $O(MKd^2 + LK^2)$. In addition, we compare *ROAM* with key baseline models and list the relevant complexity in Table 14. Compared to baseline methods such as MoMoK and OTKGE, which rely on fixed expert assignment or one-time global alignment, *ROAM* introduces a more expressive and adaptive routing process. This leads to better representation quality and improved task performance across various benchmarks. However, these improvements come at the cost of increased computational complexity. Specifically, the time complexity of *ROAM*, $O(MKd^2 + LK^2)$, is higher than that of simpler models like Sinkhorn-MoE ($O(Kd^2 + LK)$). This is primarily due to the use of multiple sub-experts and the relation-aware OT solver. **In summary**, although *ROAM* incurs greater computational overhead than some lightweight alternatives, the performance gains it delivers, particularly in terms of expert specialization, modality disentanglement, and reasoning accuracy, justify the additional cost, making it a compelling choice for applications where robustness and fine-grained reasoning are essential.

Table 14: Time complexity comparison for a single training step (per batch). Notation: $d$ dimension, $M$ modalities, $K$ experts/groups, $L$ Sinkhorn iterations, $|E|$ entity set size. We typically have $M < K < L \ll d \ll |E|$.

| Model | Time Complexity | Model | Time Complexity |
|---|---|---|---|
| ***ROAM*** | $O(MKd^2 + LK^2)$ | OTKGE | $O(MLd^2)$ |
| MoMoK | $O(MKd^2)$ | MMRNS | $O(Md^2 + |E|d)$ |
| Sinkhorn-MoE | $O(Kd^2 + LK)$ | MMKRL | $O(Md^2)$ |

# F  DISCUSSION

While *ROAM* builds upon the established concepts of Mixture-of-Experts (MoE) and Optimal Transport (OT), it is not merely a combination of existing components. As summarized in Table 1, *ROAM* represents a **conceptual shift** for handling with MMKGC. Unlike prior works that rely on unconditional pre-determined cost metrics ($\diamond, \triangle$) and export prototypes ($\heartsuit$), *ROAM* introduces orthogonal architectural choices, with relation-conditioned routing and parameter-driven group representation, to address the unique challenges of cross-modal reasoning. Below, we discuss in details how *ROAM* depart from traditional methods.

## F.1  RELATION-CONDITIONED, LEARNABLE TRANSPORT COST

Existing OT-MoE methods typically employ fixed geometric distances or relation-agnostic similarities as the transport cost. Such costs do not model relational semantics and therefore fail to align with the reasoning objective of MMKGC. In contrast, *ROAM* (ReGOR) introduces a relation-conditioned, learnable transport cost, where the cost explicitly depends on the entity–relation interaction and the functional behavior of each expert group. This yields a functional form of OT in which both sides of the transportation map correspond to functions rather than static embeddings, and can be interpreted as a Kantorovich relaxation of the Monge problem. To our knowledge, this relation-aware functional OT formulation has not been explored in existing OT–MoE literature.

## F.2  PARAMETER-DRIVEN EXPERT-GROUP REPRESENTATION

Prior MoE systems often rely on prototypes or learned embeddings to represent experts. However, prototypes cannot capture the evolving specialization of experts during training. ROAM constructs group representations directly from the parameters of their sub-experts, enabling the router to observe group behavior rather than a fixed embedding. Our ablation (Table 4) shows that replacing this parameter-driven representation with a prototype leads to consistent degradation across datasets, demonstrating that this representation is essential for robust specialization and mitigating distributional drift.

## F.3  GROUP VS EXPERT

In ROAM, an expert group is designed as a cross-modal subnetwork with a learned functional preference. Specifically, each group consists of modality-specific sub-experts (e.g., structural, textual,

visual), which are jointly optimized during training to specialize in particular types of relational and entity-level patterns. In other words, an expert group captures a particular relational–entity interaction pattern, encoding how different modalities should be integrated when reasoning.

First, a single shared expert pool (SEP) forces experts to process heterogeneous modalities jointly, which leads to severe *modality dominance*, where stronger modalities suppress weaker ones. This phenomenon has been repeatedly reported in recent studies (Yang et al., 2024; Zhang et al., 2024a; Yang et al., 2025; Zhang et al., 2024c), showing that unified routing tends to amplify modality imbalance and can even collapse into a single-modality solution. Consistent with these findings, our SEP variant yields the weakest performance across all datasets, confirming that unsegmented mixing is suboptimal.

Second, fully independent modality-specific routing (MGE) alleviates modality dominance but introduces what prior work describes as *cross-modal fragmentation*, the failure to capture complementary interactions between modalities (Zhang et al., 2024a; Yu et al., 2024). By routing each modality in isolation, MGE fails to capture the cross-modal synergies necessary for effective fusion. As such, MGE performs better than SEP but still lags behind MoSEG in our experiments.

To address both issues, our design introduces the **expert group**. Each group contains modality-specific sub-experts but performs relation-aware intra-group fusion before routing. This structure (i) *preserves modality-specific isolation* inside each group, thereby preventing dominant modalities from overwhelming others; and (ii) enables *semantic specialization across groups*, which assigns each (entity, relation) pair to the most appropriate multimodal interaction pattern.

## F.4 ROUTING STRATEGY

A unique aspect of *ROAM* is the strategic reuse of relational information ($r$) in both the expert encoding phase (MoSEG) and the routing phase (ReGOR). This is not a redundant design but reflects two functionally distinct roles crucial for reasoning:

- **Modulation Role (Inside Groups):** Within each expert group, $r$ modulates the attention weights to dynamically fuse structural, textual, and visual features. This creates a relation-aware group representation that is tailored to the specific reasoning task.
- **Decision Role (Routing Cost):** In the ReGOR module, $r$ defines the conditional transport cost $c \mid r$, which determines *which* group is selected based on semantic compatibility. By this design, *ROAM* replaces soft attention weighting with *relation-conditioned Optimal Transport routing*, where the learned cost function enables sparse and semantically aligned expert selection.

A detailed feature-by-feature comparison as shown in Table 15.

Table 15: Comparison of routing strategy with traditional methods.

| | Attention-based (MoMoK) | ROAM (Ours) |
|---|---|---|
| **Routing Mechanism** | No explicit routing; all experts blended via feature weighting. | **Optimal Transport-based routing** with a learnable, relation-conditioned cost function. |
| **Role of Relation** | Relations do not define routing destinations; instead, they only modulate attention weights (e.g., scaling or adjusting Softmax temperature). | Relations **directly shape the transport cost**, determining how entities are assigned to expert groups under the OT routing framework. |

