# OpenReview forum: "ROAM: A Relation-aware Optimal Transport-based Adaptive Mixture-of-Expert-Group Framework for Multimodal Knowledge Graph Completion"
_ICLR.cc/2026/Conference — ICLR 2026 Conference Desk Rejected Submission_

### Official Review · Reviewer_w6dR · 2025-10-26

**Soundness:** 3
**Presentation:** 2
**Contribution:** 3
**Rating:** 6
**Confidence:** 2

**Summary:**

This paper presents ROAM, a new framework for MMKGC. ROAM integrates Optimal Transport and Mixture-of-Experts architectures via relation-aware adaptive expert routing. The model consists of two main modules: MoSEG, which builds modality-specific expert groups to reduce cross-modal interference, ReGOR, a relation-guided OT routing mechanism with a learnable cost function for dynamic group selection. Extensive experiments on four MMKGC benchmarks demonstrate consistent state-of-the-art results, with up to 9.76% relative improvement, supported by ablation and efficiency analyses.

**Strengths:**

**Originality:**

1.The first work to unify OT and MoE under a relation-aware routing paradigm for MMKGC.
2.Proposes dynamic expert-group representation using learnable parameters, which is superior to the fixed prototype-based model.

**Quality:**

1.Theoretical formulation is solid and mathematically detailed.
2.Comprehensive experiments across multiple datasets and baselines with fair comparisons. Ablation, sensitivity, and efficiency analyses convincingly support the claims.

**Clarity:**
1. Well-structured presentation with clear modular separation between MoSEG and ReGOR.

**Significance:**

1. The framework achieves notable performance gains.

**Weaknesses:**

1.Although complexity is analyzed, ROAM incurs higher time and memory costs compared to some baselines (e.g., MMKRL), with limited discussion on practical deployment. And the scalability discussion is limited, it remains unclear how ROAM performs on large-scale knowledge graphs.
2.The paper lacks interpretability analysis, e.g., visualization of expert activation under different relation types.

**Questions:**

1.How robust is ROAM in scenarios with weak or noisy relational semantics?
2.Could the efficiency of ReGOR be further improved, e.g., via sparse routing or approximate OT methods?
3.Part of the font in Figure 1 is somewhat unclear and can be appropriately resized.

---

> ### Author Response · Authors · 2025-11-21
> **[Part 1/2] Response to Reviewer w6dR**
>
> We are genuinely grateful for your detailed review and accurate grasp of our contributions. Below, we present a point-by-point response to your comments and outline the revisions made to address them.
>
>
>
> > **W1** Compared to baselines,  it remains unclear how ROAM performs on large-scale knowledge graphs.
>
> We thank the reviewer for raising this concern. we conducted additional experiments on a significantly larger graph, FB15k-237-1MG (|$E$| = 14,541, |$R$| = 237, |$T$| = 310,116), which significantly exceeds the scale of our existing benchmarks (e.g., containing nearly $1.7\times$ the triples of KVC16K). The results are summarized in Table A below.
>
>
> #### Table A. Performance comparison of different models (Metric: MRR in %). Note that ( * ) denotes results are directly sourced from original papers and ( - ) denots results that are not reported in original. Baselines such as MoMoK, MMRNS, OTKGE, and MMKRL are already cited in the manuscript.*
>
> | Model          | MRR       | Hit@1   | Hit@3   | Training Time (s) |
> | :---           | :---:      | :---:      | :---:      | :---:             |
> | MoMoK          | 35.52     | 26.07     | 39.32     | 26.9     |
> | MMRNS          | 33.16     | 25.02     | 35.80     | 192.7    |
> | OTKGE          | 32.81     | 24.26     | 35.67    | 72.5     |
> | MMKRL          | 31.45     | 23.05     | 34.26     | 23.0     |
> | MEOW [1]          | 37.9 (*)  | 28.10 (*) | 39.10 (*) | 62.2 (*) |
> | DM-MKGC [2]       | 34.6 (*)  | 26.50 (*) | 37.10 (*) | (-)      |
> | **ROAM (Ours)**| **42.66** | **33.39** | **46.77** | **32.1** |
>
> Despite the increase in graph size, ROAM consistently outperforms existing KGC baselines, achieving higher MRR and Hits@K. In terms of training efficiency, ROAM also offers clear advantages: it is approximately 6× faster than MMRNS and more than 2× faster than OTKGE and MEOW, while remaining comparable to lightweight MoE-based methods such as MoMoK. These results demonstrate that ROAM scales effectively to large KGs while maintaining strong performance.
>
> These points have been included in the revised manuscript under **Appendix C.1**.
>
> [1] Multi-modal Entity in One Word: Aligning Multi-level Semantics for Multi-modal Knowledge Graph Completion, IEEE Transactions on Big Data, 2025.
>
> [2] DM-MKGC: Multimodal Knowledge Graph Completion Based on Dynamic Prompt Learning and Multi-granularity Aggregation, IEEE Transactions on Circuits and Systems for Video Technology, 2025.
>
>
>
> > **W2** The paper lacks interpretability analysis, e.g., visualization of expert activation under different relation types.
>
> We thank the reviewer for highlighting the importance of interpretability. To address this, we refer to the qualitative analyses provided in `Appendix B`, which visualize how the model adapts to different relations and modalities. For instance, `Fig. 5` visualizes the routing paths for various (entity, relation) pairs, where the model learns to associate specific expert groups with specific entity-relation types. On the other hand, `Fig. 6` shows the attention weights assigned to different modalities within each expert group, wher some prioritize textual features and others focus on structures.

---

> > ### Author Response · Authors · 2025-11-21
> > **[Part 2/2] Response to Reviewer w6dR**
> >
> > > **Q1** How robust is ROAM in scenarios with weak or noisy relational semantics?
> >
> > To evaluate the robustness of ROAM under conditions of weak or noisy semantics, a perturbation experiment is performed via injecting random Gaussian noise (10% and 20%) into the input representations. As summarized in Table B, ROAM exhibits remarkable resilience to such perturbations. Specifically, even when subjected to 20% noise, the performance degradation is minimal across all datasets; for instance, the MRR on MKG-W decreases by only 1.15% (from 40.08% to 38.93%), and on MKG-Y by merely 0.61%.
> >
> >
> > #### Table B. Robustness analysis against modal noise (Metric: MRR / Hit@1 in %).
> >
> > | Dataset | Noise-free (0%) | 10% Noise | 20% Noise |
> > | :--- | :---: | :---: | :---: |
> > | **MKG-W** | 40.09 / 34.22 | 39.33 / 33.41 | 38.94 / 33.07 |
> > | **MKG-Y** | 40.17 / 37.31 | 39.75 / 36.98 | 39.56 / 36.56 |
> > | **DB15K** | 44.19 / 36.32 | 42.51 / 34.42 | 41.85 / 33.06 |
> > | **KVC16K** | 19.89 / 12.31 | 19.23 / 11.85 | 18.97 / 11.57 |
> >
> > These points have been included in the revised manuscript under **Appendix C.3**.
> > > **Q2** Could the efficiency of ReGOR be further improved, e.g., via sparse routing or approximate OT methods?
> >
> > We thank the reviewer for the constructive suggestion regarding efficiency. To better understand the efficiency–accuracy trade-off of ReGOR, we implement three variants of the ReGOR routing module: Top-K Sparse Routing (**TKSR**), which removes the iterative OT optimization and instead directly routes each token to the $k=2$ lowest-cost expert groups based on the cost matrix[1]; Sparse Optimal Transport (**SOT**), which still solves an OT problem but enforces sparsity by masking out entries in the cost matrix whose values exceed a threshold $\tau = 0.1$[2]; and Low-rank approximate OT (**LOT**), which keeps the Sinkhorn-style OT formulation but employs a low-rank matrix factorization on the cost matrix and reduces the number of Sinkhorn iterations by 50\% (i.e., $L = 25$)[3].
> >
> >
> > #### Table C. Efficiency and performance comparison of routing strategies (Metric: MRR / Hit@1 in %).
> >
> >
> > | Dataset | Method | MRR | Hit@1 | Training (s) | Inference (s) |
> > |:---|:---|:---|:---|:---|:---|
> > | **MKG-W** | TKSR | 39.18 | 32.86 | 4.87 | 6.47 |
> > | | SOT | 39.47 | 33.15 | 4.88 | 6.49 |
> > | | LOT | 39.80 | 33.94 | 4.89 | 6.51 |
> > | | **ROAM** | **40.09** | **34.22** | **4.93** | **6.53** |
> > | | | | | | |
> > | **MKG-Y** | TKSR | 39.75 | 36.66 | 3.11 | 6.48 |
> > | | SOT | 39.84 | 36.81 | 3.12 | 6.50 |
> > | | LOT | 39.94 | 37.09 | 3.14 | 6.52 |
> > | | **ROAM** | **40.17** | **37.31** | **3.16** | **6.56** |
> > | | | | | | |
> > | **DB15K** | TKSR | 43.03 | 34.41 | 11.35 | 20.48 |
> > | | SOT | 43.90 | 35.87 | 11.36 | 20.49 |
> > | | LOT | 44.08 | 36.09 | 11.38 | 20.51 |
> > | | **ROAM** | **44.19** | **36.32** | **11.42** | **20.54** |
> > | | | | | | |
> > | **KVC16K** | TKSR | 19.25 | 11.82 | 25.65 | 27.15 |
> > | | SOT | 19.72 | 12.15 | 25.72 | 27.22 |
> > | | LOT | 19.85 | 12.28 | 25.78 | 27.28 |
> > | | **ROAM** | **19.89** | **12.31** | **25.82** | **27.32** |
> >
> > Table C summarizes the results of these routing variants across four benchmarks, reporting both performance metrics (MRR/Hits) and training/inference time relative to the full ROAM model. Although these variants offer slight speed benefits, they consistently incur performance degradation.
> >
> > As detailed in `Appendix D`, the total single-step computational complexity of ROAM is $\mathcal{O}(MKd^2 ;+; LK^2)$, where the embedding dimension ($d$) is typically much larger than the number of expert groups ($K$) and the number of Sinkhorn iterations ($L$) (i.e., $d \gg K, L$). As a result, the **encoding term** $\mathcal{O}(MKd^2)$ dominates the overall cost, while the routing term $\mathcal{O}(LK^2)$ contributes a small fraction of the total computation. Consequently, methods such as **TKSR** or **LOT**, which primarily reduce the routing complexity, have limited effect on end-to-end runtime.
> > These points have been included in the revised manuscript under **Appendix C.3**.
> >
> >
> > [1] Outrageously Large Neural Networks: The Sparsely-Gated Mixture-of-Experts Layer, ICLR, 2017.
> >
> > [2] Sparsity-Constrained Optimal Transport, International Conference on Learning Representations (ICLR), 2023.
> >
> > [3] Low-Rank Sinkhorn Factorization, International Conference on Machine Learning (ICML), 2021.
> >
> > > **Q3** Part of the font in Figure 1 is somewhat unclear and can be appropriately resized.
> >
> > We thank the reviewer for pointing this out. In the revised version, we resize the fonts, increase contrast, and adjust spacing to ensure that all labels and annotations are clearly visible.

---

### Official Review · Reviewer_dMkC · 2025-10-26

**Soundness:** 3
**Presentation:** 3
**Contribution:** 2
**Rating:** 4
**Confidence:** 3

**Summary:**

This paper addresses the challenge of multimodal information fusion in Multimodal Knowledge Graph Completion (MMKGC) by proposing ROAM, a Relation-aware Optimal-transport-based Adaptive Mixture-of-expert-group framework. The key contributions include: (1) enhancing multimodal semantic fusion through relation-aware modeling, (2) optimizing mixture-of-experts gating via Optimal Transport theory, and (3) improving modality-specific routing by dynamically representing experts based on their parameters. Experimental results demonstrate consistent improvements over existing methods across multiple benchmark datasets.

**Strengths:**

1.This work represents the first integration of Optimal Transport with Mixture-of-Experts for MMKGC, offering a novel perspective on multimodal fusion.

2.The proposed method achieves measurable performance gains across several established benchmarks, validating its practical effectiveness.

**Weaknesses:**

1.The distinction between ROAM and Sinkhorn-MoE — particularly regarding the integration of OT and MoE — remains inadequately clarified, beyond differences in task specificity.

2.Ablation studies on the ReGOR component show only marginal differences between variants, raising questions about the necessity of its individual elements.

**Questions:**

1.Could the authors elaborate on the fundamental differences between ROAM and Sinkhorn-MoE in terms of how OT and MoE are combined, beyond task-level distinctions?

2.Given the limited performance gaps observed in the ReGOR ablations, what further evidence supports the necessity of each sub-component? Are there qualitative or quantitative analyses that better demonstrate their contributions?

---

> ### Author Response · Authors · 2025-11-21
> **[Part 1/2] Response to Reviewer dMkC**
>
> We sincerely appreciate your insightful comments and the questions raised regarding our work. We have thoroughly reviewed each point and incorporated the necessary revisions and clarifications to address your concerns:
>
>
>
> > **W1&Q1**  What are the fundamental differences between ROAM and Sinkhorn-MoE regarding the integration of OT and MoE, beyond task-level distinctions?
>
> We thank the reviewer for this insightful question and the opportunity to further clarify our methodology. Beyond differences in downstream tasks, ROAM and Sinkhorn-MoE diverge in **several core methodological aspects** in how OT and MoE are integrated (`Table 1`). Below, we summarize the key distinctions.
>
>
>
> #### **1. Relation-conditioned learnable vs. unconditional predetermined OT cost**
> Sinkhorn-MoE performs routing using an **unconditional, predetermined OT cost**, where a fixed linear router computes token–expert similarities that directly serve as the transport cost. Consequently, its routing mechanism remains **agnostic to relational semantics** and cannot adapt the cost to specific inputs.
> ROAM, in contrast, introduces a **relation-conditioned, learnable OT cost**, in which the transport cost explicitly depends on the entity–relation interaction and the outcome of each expert group. *This yields a form of functional OT and can be viewed as a Kantorovich relaxation of the Monge problem (Lines 249–251).* This relation-aware, functional-cost OT formulation is fundamentally different from the unconditional, prototype-based routing used in Sinkhorn-MoE.
>
> #### **2. Functional (parameter-driven) vs. prototype-based expert representation**
> Sinkhorn-MoE represents each expert using **prototype vectors**, which cannot capture how experts specialize or evolve during training.
> ROAM instead constructs expert-group representations **directly from the parameters of their sub-experts**, allowing the router to follow the group function-level behavior. This leads to more specialization under evolving relational contexts. Our ablation (`Table 4`) confirms that replacing this parameter-driven representation with prototypes results in consistent performance degradation.
>
>
>
> Together, these design choices form a principled and methodologically distinct integration of OT and MoE, clearly differentiating ROAM from Sinkhorn-MoE.
>
> These points have been included in the revised manuscript under **Appendix F**.

---

> ### Author Response · Authors · 2025-11-21
> **[Part 2/2] Response to Reviewer dMkC**
>
> > **W2&Q2** What further evidence supports the necessity of ReGOR's sub-components given the limited performance gaps in the ReGOR ablations?
>
> We thank the reviewer for the opportunity to clarify this point. The ablation in `Table 4` evaluates the three sub-components of ReGOR *within our full functional-OT formulation*:
> $$
> c_{ik} \mid r_j = \phi\big(f(e_i, r_j) \| g(W^{(k)})\big). \tag{Eq. 6}
> $$
>
> The relatively small gaps in `Table 4` are expected because each ablation **removes only one component at a time**, while the other two still remain active. This means that even when one part is removed, the system continues to benefit from the remaining components, making the degradation appear modest. Thus, `Table 4` alone may underestimate the necessity of the individual components.
>
>
> To more clearly isolate the contribution of each sub-component, we further conduct a controlled experiment using a **conventional OT–MoE routing cost** as the **Baseline-Proto**:
> $$
> c_{ik} \mid r_j = 1 - \text{cosine}(e_i\|r_j, v_k),
> $$
> where $v_k$ is the $k$-th expert prototype (as used in Sinkhorn-MoE).
>
>
> Accordingly, we **reintroduced our three components one at a time**. The following variants are constructed: (1) *+LearnableCost* (introduces the learnable cost network):  $c_{ik} \mid r_j = \phi(e_i\|r_j\| v_k)$, (2) *+DynGroupRep* (replaces prototypes with dynamic group representation): $c_{ik} \mid r_j = 1 - \text{cosine}(e_i\|r_j , g(W^{(k)}))$,  (3) *+RelTransform* (adds relation-conditioned entity transformation): $c_{ik} \mid r_j = 1 - \text{cosine}(f(e_i, r_j),v_k)$
>
>
>
> #### Table A. Ablation study of ReGOR sub-components (Metric: MRR in %).
> | Model Variant      | MKG-W | MKG-Y | DB15K | KVC16K |
> | :---               | :---  | :---  | :---  | :---   |
> | **Baseline-Proto** | 36.42 | 36.74 | 39.22 | 18.05  |
> | **+LearnableCost** | 37.51 | 37.95 | 40.58 | 18.82  |
> | **+DynGroupRep** | 37.78 | 38.34 | 40.15 | 18.95  |
> | **+RelTransform** | 37.79 | 39.18 | 41.28 | 18.65  |
>
> The results is present in **Table A**, where each proposed component produces a noticeable gain, demonstrating that ReGOR's design elements are individually necessary.
>
> We hope this additional clarification help address the reviewer's concerns.
> These points have been included in the revised manuscript under **Appendix C.2**.

---

### Official Review · Reviewer_36XF · 2025-10-26

**Soundness:** 2
**Presentation:** 3
**Contribution:** 3
**Rating:** 4
**Confidence:** 4

**Summary:**

This paper proposes ROAM, a novel framework for Multimodal Knowledge Graph Completion that integrates Mixture-of-Experts with Optimal Transport in a relation-aware manner. ROAM organizes modality-specific experts into groups and employs a relation-guided OT routing mechanism with a learnable cost function, enabling dynamic and context-sensitive expert selection. Extensive experiments on four benchmarks demonstrate that ROAM achieves state-of-the-art performance, with significant improvements over existing methods.

**Strengths:**

S1. The paper includes rigorous ablation studies, hyperparameter analysis, and statistical significance tests, providing strong empirical support for the proposed model’s effectiveness.

S2. The paper is well-written and easy to follow. The paper is well-structured.

**Weaknesses:**

W1. The motivations for certain key technical designs remain unclear. For instance, why does the author employ expert groups? What does each group represent, and what specific information do they capture? Additionally, the relation condition is already utilized within each expert. Why is it necessary to reuse this condition during routing to incorporate relational information again?

W2. The key technical contribution of the paper remains unclear. Given that both Mixture-of-Experts and Optimal Transport are already well-established techniques in multimodal knowledge graph learning, the authors should more explicitly highlight their novel methodological elements. Furthermore, the core motivation behind this work requires further clarification. Simply stating that "no prior work has unified OT and MoE for the MMKGC task" does not sufficiently justify the contribution; a deeper rationale explaining why such a unification is beneficial or necessary should be provided.

**Questions:**

None

---

> ### Author Response · Authors · 2025-11-21
> **[Part 1/2] Response to Reviewer 36XF**
>
> We sincerely thank the reviewer for the constructive feedback and valuable insights. We have carefully revised the manuscript to address these comments and incorporate the suggested improvements. Our detailed point-by-point response is provided below:
>
> > **W1.1** Why does the author employ expert groups?
>
> We thank the reviewer for raising this point. Our design motivation for expert group is rooted in *a careful analysis of the limitations of existing MoE architectures* in multimodal fusion, and our ablation results on two variants (`SEP and MGE` in `Table 3`) directly validate these motivations. Our intention may not have been stated with sufficient clarity, and we appreciate the opportunity to make it explicit.
>
>
> First, a single shared expert pool (`SEP`) forces experts to process heterogeneous modalities jointly, which leads to severe **modality dominance**, where stronger modalities suppress weaker ones. This phenomenon has been repeatedly reported in recent studies [1,2,3,5], showing that unified routing tends to amplify modality imbalance and can even collapse into a single-modality solution. Consistent with these findings, our SEP variant yields the weakest performance across all datasets, confirming that unsegmented mixing is suboptimal.
>
> Second, fully independent modality-specific routing (`MGE`) alleviates modality dominance but introduces what prior work describes as **cross-modal fragmentation**, the failure to capture complementary interactions between modalities [2,4]. By routing each modality in isolation, MGE fails to capture the cross-modal synergies necessary for effective fusion. As such, MGE performs better than SEP but still lags behind MoSEG in our experiments.
>
> To address both issues, our design introduces **expert group**. Each group contains modality-specific sub-experts but performs relation-aware intra-group fusion before routing. This structure (i) *preserves modality-specific isolation* inside each group, thereby preventing dominant modalities from overwhelming others; and (ii) enables *semantic specialization across groups*, which assigns each (entity, relation) pair to the most appropriate multimodal interaction pattern. Empirically, MoSEG achieves the strongest improvements in `Table 3` because it *simultaneously maintains modality-specific expertise and captures cross-modal complementarity at the group level*.
>
>
> These points have been included in the revised manuscript under **Appendix F**.
>
> [1] Facilitating Multimodal Classification via Dynamically Learning Modality Gap, NeurIPS 2024.
>
> [2] Multimodal Representation Learning by Alternating Unimodal Adaptation (MLA), CVPR 2024.
>
> [3] Learning to Rebalance Multi-Modal Optimization by Adaptively Masking Subnetworks, TPAMI 2025.
>
> [4] MMoE: Enhancing Multimodal Models with Mixtures of Multimodal Interaction Experts, EMNLP 2024.
>
> [5] Unleashing the Power of Imbalanced Modality Information for Multi-modal Knowledge Graph Completion, LREC-COLING 2024.
>
> > **W1.2** What does each group represent, and what specific information do they capture?
>
> We thank the reviewer for this question. In ROAM, an expert group is designed as **a cross-modal subnetwork with a learned functional preference**. Specifically, each group consists of modality-specific sub-experts (e.g., structural, textual, visual), which are jointly optimized during training to specialize *in particular types of relational and entity-level patterns*. In other words, **an expert group captures a particular relational–entity interaction pattern**, encoding how different modalities should be integrated when reasoning. `Appendix B.1 (Fig. 5)` empirically shows clear relation-conditioned specialisation, where the model switches groups for the same entity depending on the relation, indicating fine-grained, relation-specific routing. `Appendix B.2 (Fig. 6)` also verifies that individual groups learn distinct modality preferences: some favour textual features while others rely on structural ones, demonstrating that expert groups adaptively weight modalities for different reasoning preference
>
> These points have been included in the revised manuscript under **Appendix F**.

---

> > ### Author Response · Authors · 2025-11-21
> > **[Part 2/2] Response to Reviewer 36XF**
> >
> > > **W1.3**  Why is it necessary to reuse this condition during routing to incorporate relational information again?
> >
> > We thank the reviewer for this thoughtful question. The reuse of relational information in ROAM is **not a redundant design choice**, but reflects **two functionally distinct roles** that relations play in ROAM. That is, inside each expert group, the relation $𝑟$ modulates the fusion of structural/textual/visual sub-experts (`Eq. (3)–(4)`), and the goal is to obtain a relation-aware group representation. In contrast, on the routing side, the relation $r$ is used to define the conditional OT transport cost, or $c \mid r = \phi\left(f(e,r) \|\ g(\mathcal{W})\right)$ (`Eq. (6)`), which determines which expert group is selected.
> >
> > Importantly, because $f(\cdot, \cdot)$ is a learnable MLP, forcing the routing module to *ignore* $r$ is equivalent to *pushing all relation-dependent weights toward zero*. This yields a **degenerate solution as a performance lower-bound** in which the OT transport plan becomes relation-agnostic. In other words, routing will be performed where no relational context contributes to group selection, precisely *what ROAM is designed to avoid*.
> >
> >
> > To demonstrate this empirically, we evaluated a variant (only) where $r$ is removed from the routing module over 5 runs. As shown in **Table A**, the performance drop is consistent across all datasets, confirming that removing the relation results in a less discriminative and weaker routing outcome.
> >
> >
> >
> > #### Table A. Ablation analysis of relation based routing (Metric: MRR in %).
> > | Dataset | **MKG-W**  |  **MKG-Y**  | **DB15K**  | **KVC16K**  |
> > | -------- | -------- | -------- | -------- | -------- |
> > | with $r$ | 40.09     | 40.17     | 44.19     | 19.89     |
> > | w/o $r$  | 39.33     | 39.68     | 43.77    |  19.26     |
> >
> > These points have been included in the revised manuscript under **Appendix C.4**.
> >
> >
> >
> > > **W2** How does ROAM demonstrate novel methodological contributions given that MoE and OT are already well-established techniques? Furthermore, the core motivation behind this work requires further clarification.
> >
> >
> > We thank the reviewer for raising this important question. Our motivation is to address *specific limitations* of applying existing MoE/OT methods to multimodal knowledge graph completion (MMKGC).
> >
> > #### 1. Relation-conditioned, learnable transport cost (novel OT formulation)
> > Existing OT-MoE methods typically employ *fixed geometric distances or relation-agnostic similarities* as the transport cost. Such costs do not model relational semantics and therefore fail to align with the reasoning objective of MMKGC. In contrast, ROAM (ReGOR) introduces a relation-conditioned, learnable transport cost, where the cost explicitly depends on the entity–relation interaction and the functional behavior of each expert group. This yields a functional form of OT in which both sides of the transportation map correspond to functions rather than static embeddings, and can be interpreted as a Kantorovich relaxation of the Monge problem (`Lines 249–251`). *To our knowledge, this relation-aware functional OT formulation has not been explored in existing OT–MoE literature.*
> >
> > #### 2. Parameter-driven expert-group representation (novel MoE representation design)
> > Prior MoE systems often rely on *prototypes or learned embeddings* to represent experts. However, prototypes cannot capture the evolving specialization of experts during training. *ROAM constructs group representations directly from the parameters of their sub-experts, enabling the router to observe group behavior rather than a fixed embedding.* Our ablation (`Table 4`) shows that replacing this parameter-driven representation with a prototype leads to consistent degradation across datasets, demonstrating that this representation is essential for robust specialization and mitigating distributional drift.
> >
> >
> > #### 3. Cross-modal expert groups as routing units (novel MoE architecture for MMKGC)
> >
> > Existing MoE designs often suffer from modality dominance in shared expert pools or cross-modal fragmentation in modality-specific expert pools. ROAM resolves these issues by introducing cross-modal expert groups as the fundamental routing unit. Specifically, each group contains structural/textual/visual sub-experts and performs relation-aware intra-group fusion before global routing. *This architecture enables each group to encode distinctive entity–relation interaction patterns while avoiding the functional interference of shared-pool MoEs and the cross-modal fragmentation of fully separated MoEs.*
> > These points have been included in the revised manuscript under **Appendix F**.

---

### Official Review · Reviewer_rWS6 · 2025-10-28

**Soundness:** 2
**Presentation:** 3
**Contribution:** 2
**Rating:** 6
**Confidence:** 4

**Summary:**

This paper proposes ROAM, a novel framework for Multi-Modal Knowledge Graph Completion that integrates Modality-Specific Expert Groups with a Relation-Guided Optimal Transport Routing mechanism. ROAM constructs expert group representations directly from their constituent parameters and conditions the transport cost on relational semantics. The results across multiple datasets demonstrate state-of-the-art performance of the proposed method.

**Strengths:**

The proposed ReGOR module introduces a relation-aware, learnable cost function, which is a significant advancement over prior OT-MoE methods that use fixed or prototype-based costs.

ROAM consistently outperforms existing methods across four benchmarks, with notable gains in Hits@1 and MRR.

Despite its complexity, ROAM maintains reasonable training time and GPU memory usage.

**Weaknesses:**

Table 1 is quite confusing. Does it imply that the proposed ROAM has the most powerful designed features? But I don't think more designed features equate to more strengths and advantages. While efficiency and theoretical complexity are analyzed, the scalability limits for large KGs are still a potential issue for the proposed method.

The proposed ROAM seems like a combination of existing methods. For example, MoSEG is an extension of MoMoK with group of experts, while ReGOR simply fuses the relation embedding vector into the entity embedding for computation.

**Questions:**

Please see Weaknesses.

---

> ### Author Response · Authors · 2025-11-21
> **[Part 1/2] Response to Reviewer rWS6**
>
> Thank you very much for your careful review and thoughtful comments on our manuscript. We truly appreciate your feedback and have revised the paper accordingly to fully address the concerns raised.
>
> > **W1.1** Table 1 is quite confusing. Does it imply that the proposed ROAM has the most powerful designed features?
>
>
> Thank you for pointing this out. We would like to clarify that Table 1 is not intended to suggest that ROAM benefits from having more designed features. Instead, the table highlights the **conceptual differences** across existing OT- and MoE-based methods.
>
> Specifically, ROAM *differs from prior work* by: (i) introducing a relation-conditioned, learnable OT cost (♦ + ▲), whereas existing OT-based MoE methods typically rely on unconditional, predetermined distance metrics (♢ + △); and (ii) adopting parameter-driven expert-group representations (♠) instead of prototype vectors (♡).
>
> Overall, these are not extra features, but orthogonal architectural choices that directly address core limitations of current OT-MoE approaches. These design choices enable more context-aligned and semantically grounded expert-group selection.
>
> These points have been included in the revised manuscript under **Appendix F**.
>
> > **W1.2** While efficiency and theoretical complexity are analyzed, the scalability limits for large KGs are still a potential issue for the proposed method.
>
>
> We appreciate the reviewer's careful reading of our efficiency and theoretical complexity analysis, and we fully agree that scalability is a crucial factor for all MoE methods.
>
> As stated in `Line 878`, the primary computational cost of ROAM arises from expert encoding and OT-based routing, yielding an overall complexity of $\mathcal{O}(MKd^2+LK^2)$, where $M$ is the number of modalities, $K$ the number of expert groups, $d$ the embedding dimension, and $L$ the number of Sinkhorn iterations. We emphasize that this is *on par with, and in some cases more efficient* than, several existing OT/MoE routing methods, including OTKGE, MoMoK, and MMRNS. Furthermore, `Fig. 4` empirically demonstrates this.
>
>
>
> To further address the reviewer's concern, we conduct additional experiments on **a substantially larger MMKG**, FB15k-237-1MG (|$E$| = 14,541, |$R$| = 237, |$T$| = 310,116), which significantly exceeds the scale of our existing benchmarks (e.g., containing nearly $1.7\times$ the triples of KVC16K). The results in **Table A** show that ROAM maintains competitive training time and achieve notable improvements.
>
>
> #### Table A. Performance comparison of different models (Metric: MRR in %). Note that ( * ) denotes results are directly sourced from original papers and ( - ) denotes results that are not reported in original. Baselines such as MoMoK, MMRNS, OTKGE, and MMKRL are already cited in the manuscript.*
>
> | Model          | MRR (%)      | Hit@1 (%)  | Hit@3 (%)  | Training Time (s) |
> | :---           | :---:      | :---:      | :---:      | :---:             |
> | MoMoK          | 35.52     | 26.07     | 39.32     | 26.9     |
> | MMRNS          | 33.16     | 25.02     | 35.80     | 192.7    |
> | OTKGE          | 32.81     | 24.26     | 35.67     | 72.5     |
> | MMKRL          | 31.45     | 23.05     | 34.26     | 23.0     |
> | MEOW [1]          | 37.9 (*)  | 28.10 (*) | 39.10 (*) | 62.2 (*) |
> | DM-MKGC [2]       | 34.6 (*)  | 26.50 (*) | 37.10 (*) | -       |
> | **ROAM (Ours)**| **42.66** | **33.39** | **46.77** | **32.1** |
>
> Despite the increase in graph size, ROAM consistently outperforms existing KGC baselines, achieving higher MRR and Hits@K. In terms of training efficiency, ROAM also offers clear advantages: it is approximately 6× faster than MMRNS and more than 2× faster than OTKGE and MEOW, while remaining comparable to lightweight MoE-based methods such as MoMoK. These results demonstrate that ROAM scales effectively to large KGs while maintaining strong performance. These points have been included in the revised manuscript under **Appendix C.1**.
>
>
> [1] Multi-modal Entity in One Word: Aligning Multi-level Semantics for Multi-modal Knowledge Graph Completion, IEEE Transactions on Big Data, 2025.
>
> [2] DM-MKGC: Multimodal Knowledge Graph Completion Based on Dynamic Prompt Learning and Multi-granularity Aggregation, IEEE Transactions on Circuits and Systems for Video Technology, 2025.

---

> ### Author Response · Authors · 2025-11-21
> **[Part 2/2] Response to Reviewer rWS6**
>
> > W2 The proposed ROAM seems like a combination of existing methods. For example, MoSEG is an extension of MoMoK with group of experts, while ReGOR simply fuses the relation embedding vector into the entity embedding for computation.
>
> We respectfully acknowledge the reviewer's perspective but wish to clarify that *ROAM is not merely a combination of existing techniques*. While it builds upon the general ideas of MoE and OT, the proposed MoSEG and ReGOR modules introduce *fundamentally different architectural principles* from prior work, such as MoMoK, in terms of *structural difference*, and *routing strategy*. To clearly demonstrate these distinctions, we provide a detailed feature-by-feature comparison as shown in **Table B&C**.
>
> #### Table B. Structural difference.
>
> |    | MoMoK  | ROAM |
> |:---|:---|:---|
> | **Expert organization** | Each modality maintains its own independent expert pool. | Each expert group is a multi-modality composite unit containing modality-specific sub-experts. |
> | **Input to experts** | Experts receive only entity features (no relations involved). | Relation embeddings are injected within each expert and directly participate in relation-aware attention and aggregation. |
> | **Inference** | All experts contribute and are softly aggregated using Softmax. | OT-routing solves for a sparse, relation-conditioned transport plan, selecting only relevant expert groups for inference. |
>
>
> #### Table C. Routing strategy.
> |  | MoMoK | ROAM |
> |:---|:---|:---|
> | **Routing Mechanism** | No explicit routing, as all experts are blended through attention-based feature weighting. | Optimal Transport–based routing with a learnable, relation-conditioned cost function. |
> | **Role of Relation** | Relations do not define routing destinations; instead, they only modulate attention weights (e.g., scaling or adjusting Softmax temperature). | Relations directly shape the transport cost, determining how entities are assigned to expert groups under the OT routing framework. |
>
>
> In summary, ROAM organizes *modality-specific sub-experts into multi-modality expert groups* and *injects relational context* directly into expert computation. Moreover, ROAM replaces soft attention weighting with *relation-conditioned Optimal Transport routing*, where a learned cost function enables sparse and semantically aligned expert selection.

---

### Author Response · Authors · 2025-11-27
**General response to all reviewers**

We sincerely thank all reviewers (**rWS6, 36XF, dMkC, w6dR**) for their thoughtful feedback and constructive suggestions. We are also greatly encouraged that the reviewers appreciate the *theoretical contributions and extensive experimental validation* of our method. Your expertise significantly helps us strengthen the quality and clarity of the manuscript. In addition to addressing individual comments point by point, we make substantial updates to the manuscript, as **summarized** below.

* **Scalability to large-scale KGs (Appendix C.1)**
  To address concerns regarding scalability (Reviewers **rWS6, w6dR**), we conduct additional experiments on the large-scale **FB15k-237-IMG** dataset ($|\mathcal{T}| > 310k$, containing nearly $1.7\times$ the triples of the original applied KG). The results demonstrate that ROAM scales effectively, outperforming baselines while maintaining high training efficiency.

* **Component analysis (Appendix C.2)**
  To clarify the contribution of ReGOR's sub-components (Reviewer **dMkC**), we perform additional experiments that start from a conventional baseline and add components one by one, clearly showing the significant performance gains attributed to each design element.

* **Efficiency and robustness (Appendix C.3)**

  * **Efficiency:** We evaluate three additional routing OT-based variants (TKSR, SOT, LOT) to analyze the efficiency–accuracy trade-off (Reviewer **w6dR**).
  * **Robustness:** We conduct noise perturbation experiments (10% and 20% noise) to demonstrate ROAM's resilience to noisy semantics (Reviewer **w6dR**).

* **Relational-based routing (Appendix C.4)**
  We add a detailed ablation and theoretical discussion to clarify the **role** of relational information: acting as a *modulator* inside expert groups and as a *decision factor* in the routing (Reviewer **36XF**).

* **Methodological clarifications (Appendix F)**
  We significantly expand the discussion to clarify the conceptual distinctions between ROAM and prior works (e.g., MoMoK, Sinkhorn-MoE). This includes:

  * the motivation for **Expert Groups** (solving modality dominance and fragmentation),
  * detailed comparison tables regarding **structural differences** and **routing strategies**.

* **Other**

  * We improve the readability of **Figure 1** by resizing fonts and adjusting layout (Reviewer **w6dR**).
  * We carefully correct typos and improve the presentation throughout the manuscript.

We hope these revisions comprehensively address the concerns raised. We look forward to further discussion.

Best regards,

Authors

---

### Author Response · Authors · 2025-12-03

**Dear AC, SAC and PCs,**

Thank you for handling our submission and coordinating the manuscript. Since the reviews are now frozen and there has been no follow-up after our detailed response on Nov 21, 2025, we provide a brief map from each reviewer’s main concerns to the corresponding revisions in the latest manuscript.


**[Reviewer rWS6]**

**Concerns:** scalability to large KGs; interpretation of **Table 1**; whether ROAM is just a combination of existing methods.

**Actions:** We added a larger scalability study (lines **890–913**) and clarified the role of **Table 1** and our differences from prior OT/MoE models (lines **1102–1180**).

**[Reviewer 36XF]**

**Concerns:** motivation and semantics of expert groups; reuse of relations in routing; novelty beyond “MoE + OT”.

**Actions:** We clarified the behavior of expert groups (lines **1130–1154**), the dual role of relations (lines **1005–1031**), and our methodological novelty compared with existing OT/MoE approaches (lines **1102–1180**).

**[Reviewer dMkC]**

**Concerns:** differences from Sinkhorn-MoE; necessity of ReGOR sub-components.

**Actions:** We made the contrast to Sinkhorn-MoE explicit (lines **1102–1180**) and provided a stepwise ablation showing that each ReGOR component contributes clear gains (lines **915–946**).

**[Reviewer w6dR]**

**Concerns:** training/inference cost and scalability; interpretability; robustness to noisy semantics; routing efficiency; readability of `Fig. 1`.

**Actions:** We quantified cost and scalability (lines **890–913**), evaluated more efficient and robust routing variants (lines **947–972**), and improved interpretability analyses and `Fig. 1` (**Appendix B.1–B.2**).

### [Final remark]

We have uploaded the revised version with all major changes clearly marked. We believe that our rebuttal and additional experiments have fully addressed the reviewers' concerns. Thank all reviewers for their constructive and insightful comments, which have greatly strengthened the manuscript. Please don't hesitate to let us know if you have any further questions or suggestions.

Sincerely,

The Authors

---

### Note · Program_Chairs · 2026-01-17
**Submission Desk Rejected by Program Chairs**

The following references in this submission do not refer to real documents and/or have major errors in bibliographic information:

 Yutong Liu, Xinyu Chen, Peng Zhang, Kaiyuan Xu, and Xiaolong Wang. Fedotp: Federated optimal transport with sparsity-constrained mixture-of-experts. In International Conference on Learning Representations (ICLR), 2024.
Wei Zhang, Ziyu Huang, Jing Sun, Lianmin Wang, and Di He. Adaptive expert routing for vision mixture-of-experts via entropy-regularized optimal transport. In Proceedings of the International Conference on Learning Representations (ICLR), 2023.